# Acinar to β-like cell conversion through inhibition of focal adhesion kinase

Shakti Dahiya [1,10] ✉, Mohamed Saleh[2,10], Uylissa A. Rodriguez[1,10], Dhivyaa Rajasundaram[2], Jorge R. Arbujas[1], Arian Hajihassani[1], Kaiyuan Yang [3,4], Anuradha Sehrawat[1], Ranjeet Kalsi[1], Shiho Yoshida[1], Krishna Prasadan[1], Heiko Lickert [3,4,5], Jing Hu[6], Jon D. Piganelli[1], George K. Gittes[1] & Farzad Esni [1,5,7,8,9] ✉

Insufficient functional β-cell mass causes diabetes; however, an effective cell replacement therapy for curing diabetes is currently not available. Reprogramming of acinar cells toward functional insulin-producing cells would offer an abundant and autologous source of insulin-producing cells. Our lineage tracing studies along with transcriptomic characterization demonstrate that treatment of adult mice with a small molecule that specifically inhibits kinase activity of focal adhesion kinase results in trans-differentiation of a subset of peri-islet acinar cells into insulin producing β-like cells. The acinar-derived insulin-producing cells infiltrate the pre-existing endocrine islets, partially restore β-cell mass, and significantly improve glucose homeostasis in diabetic mice. These findings provide evidence that inhibition of the kinase activity of focal adhesion kinase can convert acinar cells into insulin-producing cells and could offer a promising strategy for treating diabetes.

When β-cell depletion outpaces β-cell generation, the overall number of insulin-producing cells decreases, and a shortage of insulin becomes evident. The generation of insulin-producing cells to compensate for their absolute or relative shortage in type 1 (T1D) and type 2 (T2D) diabetes is an appropriate therapeutic strategy. Thus, a cure for diabetes should entail replacement of insulin-producing β-cells. One approach is islet transplantation, a technique that during the past four decades has evolved into a routine clinical procedure with predictable efficacy for selected T1D patients[1]. In addition to islet transplantation, there have been tremendous efforts throughout the years to generate new β-cells either through proliferation of pre-existing β-cells[2–4] or by neogenesis using different sources such as embryonic stem cells[5], duct cells[6–11], non-β-cells residing in the endocrine islets[12–15]. Another

potential treatment for this disease would be the direct conversion of pancreatic acinar cells into β-cells in sufficient numbers to restore and maintain normal concentrations of glucose in the blood. Human and rodent acinar cells can be induced in vitro to express insulin through viral-mediated expression of MAPK and STAT3[16]. Furthermore, transduction of mouse acinar cells in vivo with vectors encoding three transcription factors that are necessary for β-cell development can induce conversion of acinar cells to functional β-cells[17–19]. However, the long-term persistence of acinar-derived β-like cells requires reprogramming of a significant number of acinar cells[18]. Thus, the key to successful reprogramming would be to create a large number of newly formed β-like cells, which would enable them to form islet structures and their own niche environment[18].

[1]Department of Surgery, Division of Pediatric General and Thoracic Surgery, Children's Hospital of Pittsburgh, University of Pittsburgh Medical Center, Pittsburgh, PA, USA. [2]Department of Pediatrics, Children's Hospital of Pittsburgh, University of Pittsburgh Medical Center, Pittsburgh, PA, USA. [3]Institute of Diabetes and Regeneration Research, Helmholtz Munich, Neuherberg, Germany. [4]German Center for Diabetes Research (DZD), Neuherberg, Germany. [5]School of Medicine, Technical University of Munich, Munich, Germany. [6]Department of Medicine, Division of Gastroenterology, Hepatology and Nutrition, University of Pittsburgh, Pittsburgh, PA, USA. [7]Department of Developmental Biology, University of Pittsburgh, Pittsburgh, PA, USA. [8]UPMC Hillman Cancer Center, Pittsburgh, PA, USA. [9]McGowan Institute for Regenerative Medicine, University of Pittsburgh, Pittsburgh, PA, USA. [10]These authors contributed equally: Shakti Dahiya, Mohamed Saleh, Uylissa A. Rodriguez. ✉e-mail: Shd156@pitt.edu; farzad.esni@chp.edu

Focal adhesion kinase (FAK) is a cytoplasmic non-receptor tyrosine kinase, which is involved in mediating integrin signaling[20-22]. Cell surface integrins engage with the extracellular matrix and recruit FAK to form dynamic structures known as focal adhesions[23]. In the developing mouse pancreas, FAK activity plays a pivotal role in the branching of the epithelium, as well as specification of lineages[24,25]. In a recent report, we demonstrated that FAK controls the onset of pancreatic lineage commitment by preventing recruitment of the endothelial cells to the pancreatic epithelium[26]. Here, we provide evidence that treatment of adult mice with a compound that specifically inhibits kinase activity of phospho-FAK (pFAK) results in a remarkable trans-differentiation of a subset of peri-islet acinar cells into insulin-producing β-like cells. More importantly, the acinar-derived insulin-producing cells infiltrate the pre-existing endocrine islets and ameliorate blood glucose in diabetic mice.

## Results

### Inhibition of FAK leads to acinar lineage conversion to insulin-producing cells

We have previously reported that while conditional deletion of *Ptk2* (the gene encoding FAK) during pancreatic development in PdxCre;*Fak*^fl/fl^ mice caused delayed acinar differentiation, the pancreas of the postnatal heterozygous littermates contained cells co-expressing insulin and amylase[26]. *Pdx1* is one of the earliest markers for pancreatic lineage, thus the appearance of these insulin⁺/amylase⁺ cells in 1-month-old PdxCre;*Fak*^fl/wt^ pancreas could be due to abnormal pancreatic development in the mutant embryos. Alternatively, these insulin⁺/amylase⁺ cells could be cells in transition during either β-cell to acinar or acinar to β-cell conversion. To discriminate between these possibilities, we first treated 8-week-old wild-type C57BL/6 mice with the FAK inhibitor PF562271 (FAKi), which specifically inhibits pFAK kinase activity[27-30]. The mice received FAKi treatment (50 mg/kg) or vehicle via oral gavage twice a day for 3 weeks. The regiment used in our study was based on a phase I trial in cancer patients as well as previous studies in mice[29,30]. The mice were euthanized immediately following the 3 weeks treatment. Serum amylase and lipase, glucose tolerance and body weights were not different between the FAKi-treated and vehicle-treated cohorts (Fig. 1a–e). The pancreas of FAKi-treated mice displayed overall normal histology.

However, upon closer examination, we could occasionally find cells co-expressing insulin and amylase in the FAKi-treated pancreas (Supplementary Fig. 1a). These insulin⁺/amylase⁺ cells, which resembled those found in the PdxCre;*Fak*^fl/wt^ pancreas[26] displayed relatively lower levels of insulin and amylase compared to the regular acinar or β-cells (Supplementary Fig. 1b–d). Anecdotally, we could also detect few single endocrine cells adjacent to islets (Fig. 1f). Presence of peri-insular endocrine cells was determined by immunostaining of insulin (Fig. 1g). The normal serum pancreatic enzymes, glucose tolerance test and overall pancreas histology suggest that FAKi treatment does not have any adverse effects on the exocrine enzyme secretion nor endocrine function of the pancreas.

To determine whether chemical inhibition of FAK could convert acinar cells into insulin-producing cells, we next employed ElaCreERT2;R26^Tom^ mice (Ela^Tom^ mice), which uses the elastase promoter to target Cre-recombinase expression to pancreatic acinar cells[7,31,32]. Tamoxifen treatment leads to tomato-red expression in essentially all acinar cells in the Ela^Tom^ mice. Hence, the presence of tomato-labeled non-acinar cells would imply that these cells originated from pre-existing labeled acinar cells. In this study, 6-week-old Ela^Tom^ mice received tamoxifen via gavage feeding. At 8 weeks of age, these mice were then treated with FAKi or vehicle for 3 weeks, in conjunction with BrdU administration via drinking water. Finally, mice were sacrificed 2 weeks after cessation of the FAKi treatment. Unlike the vehicle-treated cohort, in the FAKi-treated Ela^Tom^ mice we could occasionally find Tom⁺/Ins⁺ cells outside the islets (Fig. 1h and Supplementary

Fig. 2a–d). These Tom⁺/Ins⁺ cells could be found in larger (Fig. 1h and Supplementary Fig. 2a, b) or smaller clusters (Supplementary Fig. 2c) or as single cells (Supplementary Fig. 2d). These Tom⁺/Ins⁺ cells displayed different degrees of maturation, as evident by differences in detectable GLUT2 (Supplementary Fig. 2a), a marker for β-cell maturation[33]. Moreover, amylase expression was lost in the Tom⁺/Ins⁺ cells (Supplementary Fig. 2a–d), which is consistent with the notion that lineage conversion requires cells to initially lose their identity before acquiring the new one. The presence of Tom⁺/Ins⁺ cells in the FAKi-treated mice shows a progenitor-progeny relationship between the acinar cells and insulin-producing cells.

### Acinar-derived insulin-producing (ADIP) cells infiltrate pre-existing endocrine islets

In line with previous reports[7,31], tamoxifen-treatment in the control Ela^Tom^ mice led to tomato expression in the acinar cells, whereas endocrine and duct cells were negative for tomato-labeled cells (Fig. 1i, k). However, in the FAKi-treated mice, we could find Tom⁺ cells in 40.2% ± 15.6 of endocrine islets (Fig. 1j, l–n). Among islets containing Tom⁺ cells, 35.8% ± 9.6 had less than 5 Tom⁺ cells, 57.2% ± 8.5 had 5–10 Tom⁺ cells and 6.9% ± 3.3 contained more than 10 Tom⁺ cells per islet (Fig. 1o). While we could not find any Tom⁺/Gcg⁺ cells in the FAKi-treated Ela^Tom^ mice (Fig. 1j), 76.5% ± 4.3 of Tom⁺ cells found in islets were insulin⁺ (Fig. 1p). In total, the intra-islet ADIP cells contributed to 1.85% ± 0.6 of all β-cells in the FAKi-treated mice (Fig. 1q).

Similar to the Tom⁺/Ins⁺ cells detected outside the islets, the Tom⁺/Ins⁺ cells within the islets did not express amylase (Fig. 2a). However, unlike their extra-islet counterparts where GLUT2 displayed a heterogenous expression (Supplementary Fig. 2a), the intra-islet Tom⁺/Ins⁺ cells were all positive for GLUT2 (Fig. 2b), indicating that the intra-islet ADIP cells are likely more mature than extra-islet ADIP cells. Along with GLUT2, we could find other bona fide β-cell markers such as PDX1 (Fig. 2c), and NKX6.1 in ADIP cells (Fig. 2d–f and Supplementary Fig. 2e). Consistent with our quantification analysis (Fig. 1p), confocal immunofluorescence imaging further revealed that the intra-islet Tom⁺ population was not homogenous, but rather consisted of cells at different stages in the transformation process of acinar into insulin-producing cells (Fig. 2d–f). Accordingly, we could find Tom⁺/Nkx6.1⁻/Ins⁻ cells (Fig. 2d) Tom⁺/Nkx6.1⁺/Ins⁺ cells (Fig. 2e) and Tom⁺/Nkx6.1⁺/Ins⁻ cells (Fig. 2f) embedded within the same islet. Notably, none of the intra-islet Tom⁺ cells displayed BrdU incorporation (Fig. 2g), suggesting that this phenomenon relies primarily on acinar conversion rather than proliferation of ADIP cells.

Together, our lineage-tracing data clearly shows that specific inactivation of FAK kinase activity converts acinar cells into insulin-expressing cells. Noteworthy, these cells can either form clusters outside the islets or infiltrate pre-existing endocrine islets.

### ADIP cells partially improve blood glucose in diabetic mice

FAKi-induced conversion of acinar cells into insulin-producing cells is indeed a remarkable finding. However, there is no translational benefit if these cells are not functional. If being integrated into pre-existing islets would be a key element for ADIP cells to be functional, then near-total ablation of β-cells with high dose streptozotocin (STZ) would most likely undermine the ability of ADIP cells to infiltrate the islets and restore normoglycemia. Therefore, to achieve partial β-cell ablation, we treated 10-week-old wild-type CD1 female mice with low dose (40 mg/kg) STZ. While this approach resulted in elevated random blood glucose (BG) within a week in all treated mice, only 50% developed hyperglycemia in the following weeks (Supplementary Fig. 3a). Noteworthy, mice that maintained normoglycemia never displayed BG above 300 mg/dl (Supplementary Fig. 3f–j, green line) and those who reached 300 mg/dl did not show any signs of recovery (Supplementary Fig. 3b–e, k, red line). Thus, in this cohort, a BG of 300 mg/dl appears to be a critical threshold for further progression to diabetes.

To test whether FAKi treatment would improve or reverse hyperglycemia, we next treated 10-week-old CD1 female mice with low-dose STZ. Mice that reached a random BG of 300 mg/dl were either treated with 50 mg/kg FAKi twice daily for 3 weeks (Fig. 3a, blue line) or vehicle (Fig. 3a, red line). Mice that did not reach the 300 mg/dl BG threshold were left untreated (Fig. 3a, green line). As demonstrated in Fig. 3a, FAKi treatment led to partial improvement of the hyperglycemia (Fig. 3a, blue line and Supplementary Fig. 4a). Interestingly,

one week into FAKi-treatment the random BG dropped sharply below 300 mg/dl in 20% (2/10) of mice (Supplementary Fig. 4b, c). In addition, 70% (7/10) showed a gradual decrease below 300 mg/dl post-FAKi treatment (Supplementary Fig. 4d–j), whereas 10% (1/10) did not respond and progressed to diabetes (Supplementary Fig. 4k). Importantly, mice that recovered (9/10) showed BG values below 300 mg/dl for the remainder of the study (1 month after FAKi-treatment cessation), at which time the mice were sacrificed. These data indicate that

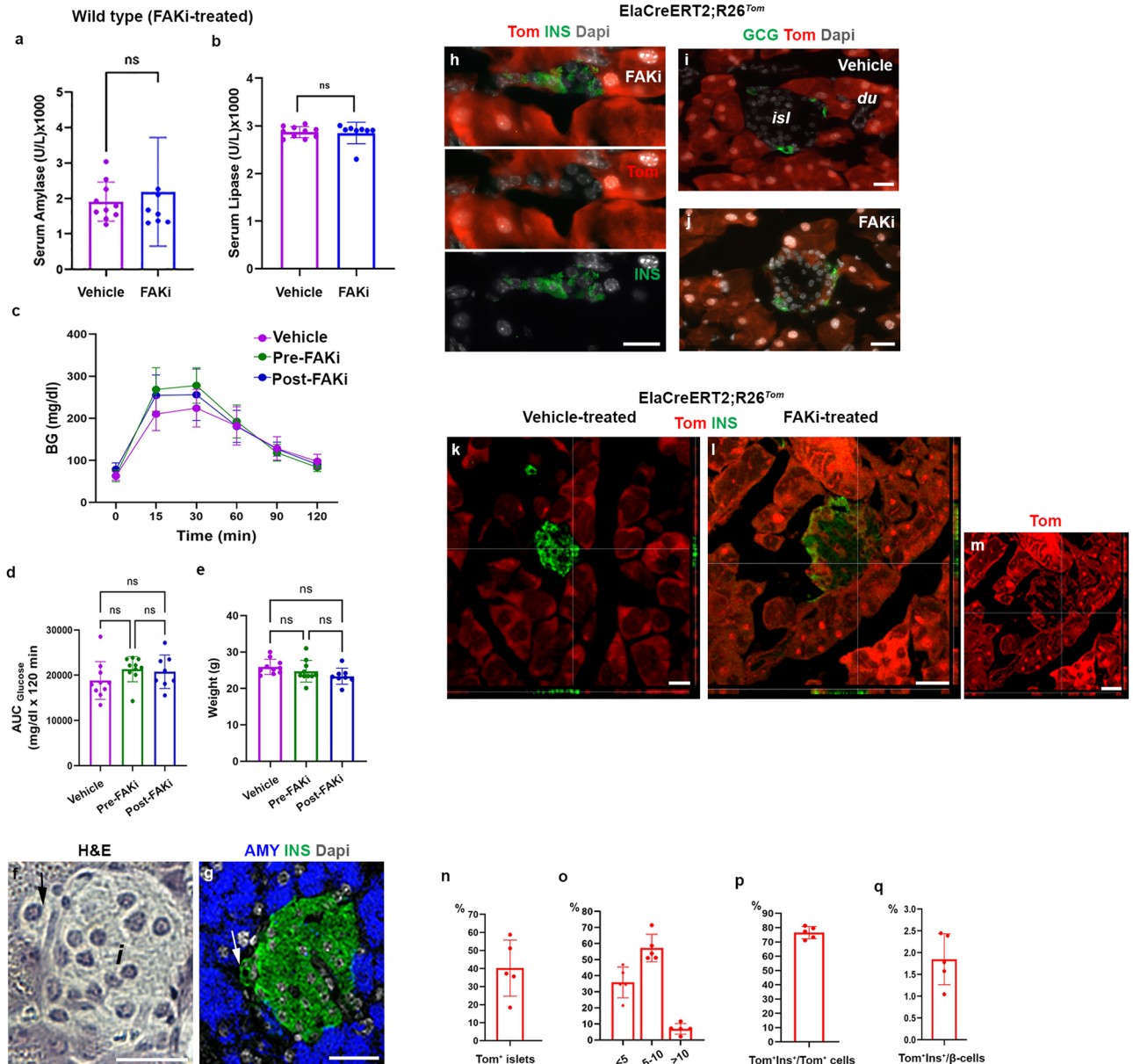

**Fig. 1 | Inhibition of FAK leads to acinar lineage conversion to insulin-producing cells. a, b** Measurement of serum amylase (**a**), or serum lipase (**b**) in vehicle-treated (*n* = 8) or FAKi-treated mice (*n* = 7 in **a**, and *n* = 8 in **b**). **c–e** Graphs showing glucose tolerance test (*n* = 10 for Vehicle, 10 for Pre-FAKi, and 8 for Post-FAKi group) (**c**), area under the curve (AUC) analysis for the IPGTT (**d**), or body weight of the mice at the time of the IPGTT (**e**). **f, g** H&E (**f**) and the corresponding immunostaining for detection of insulin and amylase on samples obtained from FAKi-treated mice showing a single insulin⁺ cell adjacent to an islet (arrows in **f** and **g**). *n* = 5 mice. **h** Imaging of tamoxifen-induced ElaCreERT2,R26^Tom mice treated with FAKi showing Tom⁺/insulin⁺ cells outside the endocrine islets. *n* = 5 mice from 3 independent experiments. **i, j** Fluorescent imaging of tamoxifen-induced ElaCreERT2,R26^Tom mice treated with vehicle (**i**) or FAKi (**j**) for detection of tomato and glucagon showing Tom⁺/Gcg⁻ cells within the islets. *n* = 5 mice from 3

independent experiments. **k–m** Confocal imaging of tamoxifen-induced ElaCreERT2,R26^Tom mice treated with vehicle (**k**) or FAKi (**l, m**) for detection of Tomato and insulin showing Tom⁺/insulin⁺ cells within islets. *n* = 5 mice from 3 independent experiments. **n–q** Quantification of tomato-labeled (Tom⁺) cells infiltrating the endocrine islets at the end of FAKi treatment. The percentage of islets infiltrated by Tom⁺ cells (**n**). The percentage of islets containing, less than 5 Tom⁺ cells, 5–10 Tom⁺ cells or more than 10 Tom⁺ cells per islet (**o**). The percentage of insulin-producing cells among the intra-islet Tom⁺ cells (**p**). The contribution of acinar-derived insulin-producing cells to the total β-cell population (**q**). *n* = 5 mice. Statistical analysis was performed using unpaired, two-tailed t-test in (**a**, **b**) and one-way ANOVA in (**d**, **e**). Data is presented as mean ± SD. *isl*: islet, *du*: duct. Scale bar = 20 μm. Source data are provided as a Source data file.

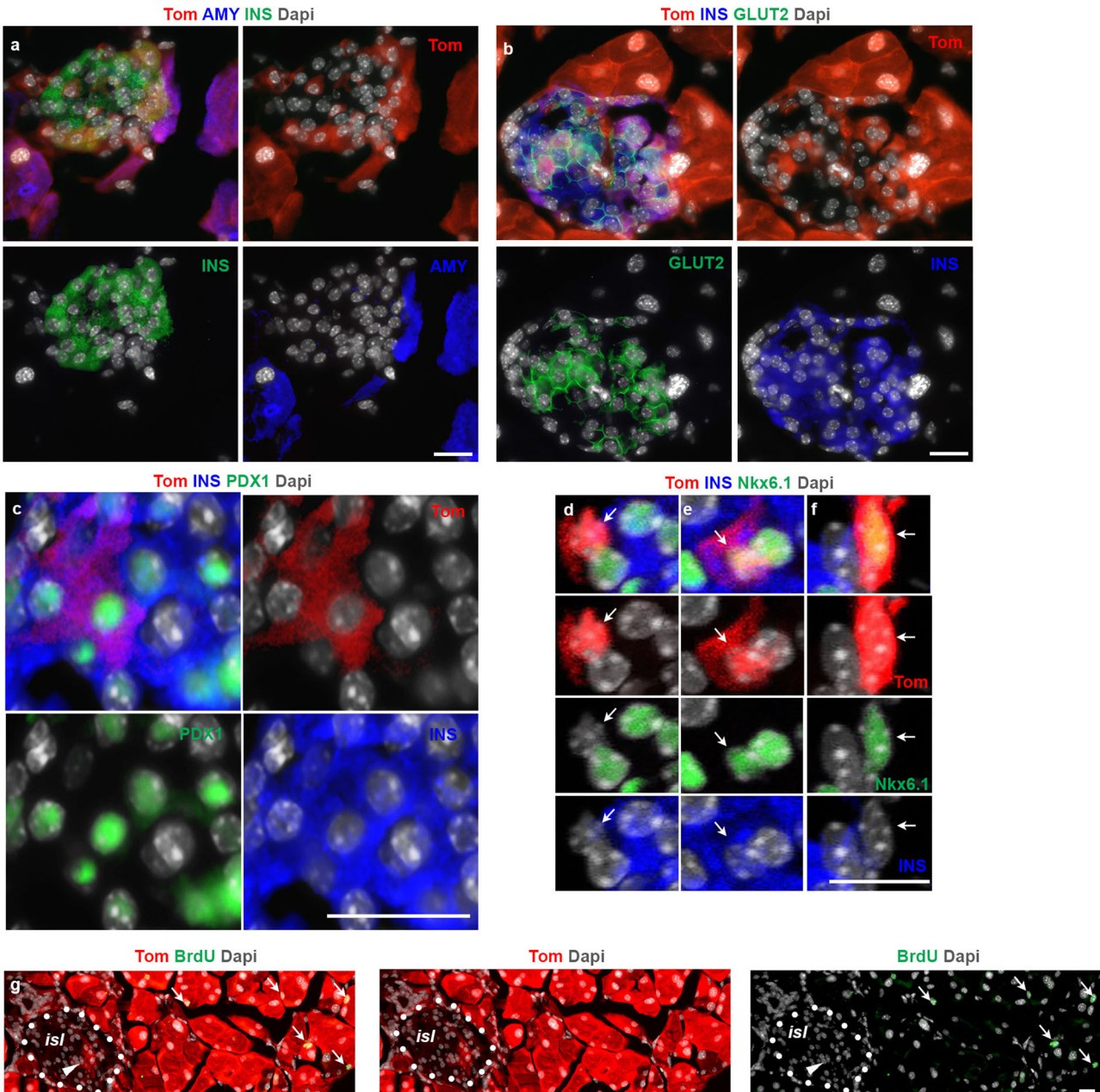

**Fig. 2 | Acinar-derived insulin-producing (ADIP) cells infiltrate pre-existing endocrine islets. a–f** Imaging of tamoxifen-induced ElaCreERT2,R26$^{Tom}$ mice treated with FAKi showing Tom$^+$/insulin$^+$ cells within the endocrine islets. $n = 5$ mice from 3 independent experiments. Fluorescent imaging of Tomato in conjunction with insulin and amylase (**a**), insulin and GLUT2 (**b**), insulin, and PDX1 showing islet infiltrated with three Tom$^+$ cells (**c**). Confocal fluorescent imaging of Tomato in conjunction with insulin and Nkx6.1 (**d–f**). **d–f** Higher magnification of three Tom$^+$ cells shown in (Supplementary Fig. 2e). Arrows highlight a Tom$^+$/Nkx6.1$^-$/Ins$^-$ cell (**d**), a Tom$^+$/Nkx6.1$^+$/Ins$^+$ cell (**e**) and a Tom$^+$/Nkx6.1$^+$/Ins$^-$ cell (**f**). **g** Fluorescent imaging of Tomato in conjunction with BrdU on tissues obtained from tamoxifen-induced ElaCreERT2,R26$^{Tom}$ mice treated with FAKi showing the absence of proliferation among Tom$^+$ cells within the endocrine islets. Dotted line mark and islet. Arrowheads in (**g**) highlight Tom$^+$ cells inside the islet. Arrows in (**g**) show BrdU$^+$ acinar cells. *isl*: islet. Scale bar = 20 μm. $n = 5$ mice from 3 independent experiments.

maintaining blood glucose does not rely on continuous FAKi-treatment.

To confirm that the observed improvement of hyperglycemia in diabetic mice is the result of acinar cell conversion (ADIP cells), we studied the effect of FAKi treatment on diabetic Ela$^{Tom}$ mice. Here, low-dose STZ-treated Ela$^{Tom}$ mice were given FAKi or vehicle once BG reached 300 mg/dl. In line with our previous results in wild-type mice treated with low dose STZ (Fig. 3a), we observed improved BG values following FAKi-treatment, whereas the vehicle-treated mice remained hyperglycemic (Fig. 3b). The mice were then euthanized either 1 week into FAKi treatment (early group) (Fig. 3c) or 4 weeks after treatment

cessation (late group) (Fig. 3d). Similar to the non-diabetic FAKi-treated mice (Fig. 2a), in both cohorts the intra-islet Tom$^+$ cells had lost amylase expression and could be detected among pre-existing β-cells that had survived the STZ-treatment (Fig. 3c, d). However, while the Tom$^+$ cells in the early cohort did not express insulin (Fig. 3c), the Tom$^+$ cells in the late cohort displayed a mixture of insulin$^+$ and insulin$^-$ cells, suggesting a progressive transition of acinar cells into insulin-producing cells (Fig. 3d). The appearance of Tom$^+$/Ins$^+$ cells post-FAKi treatment (Fig. 3d) is consistent with the observed improvement in hyperglycemia in diabetic mice treated with FAKi (Fig. 3a and Supplementary Fig. 4).

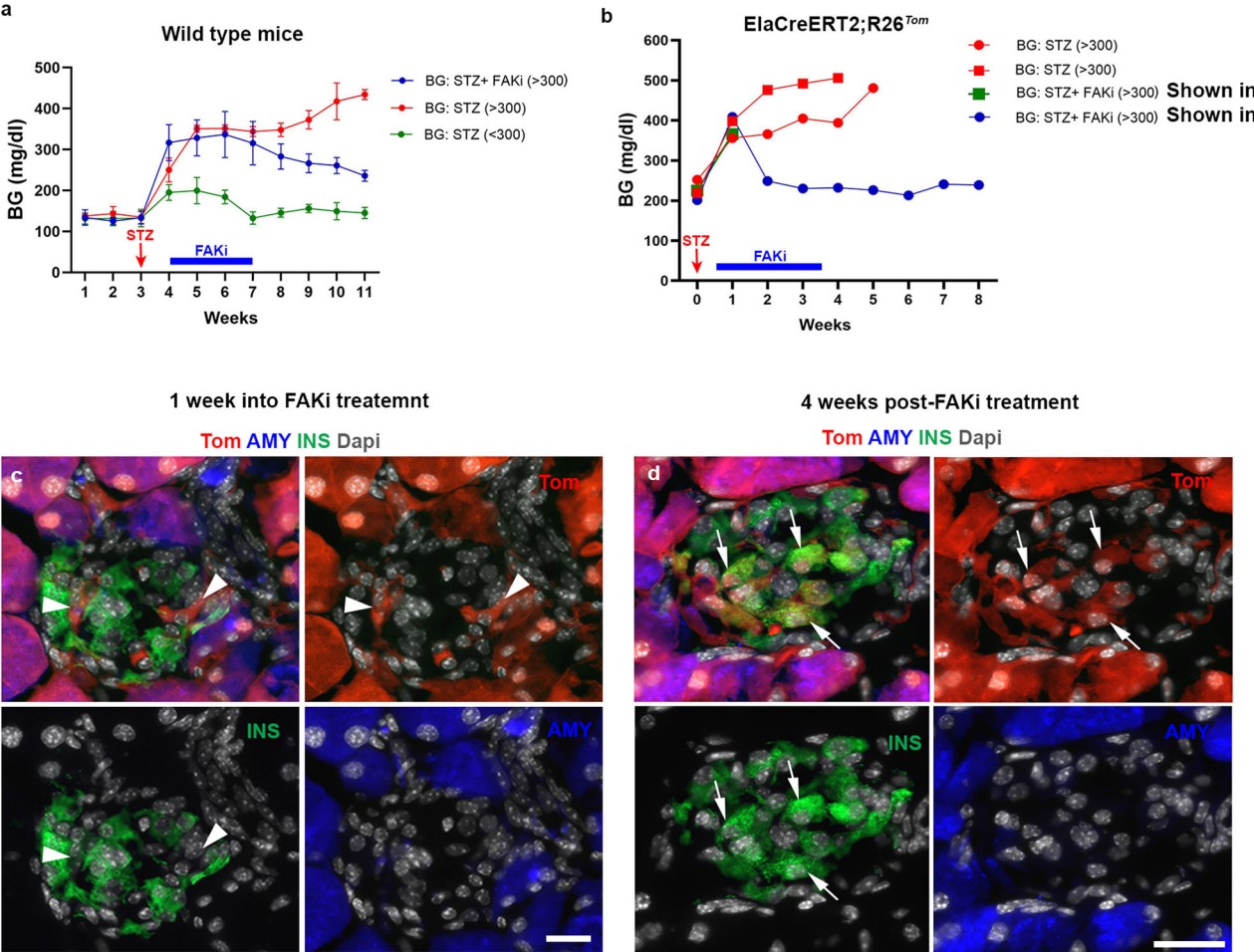

**Fig. 3 | ADIP cells ameliorate blood glucose homeostasis in mice following low dose streptozotocin treatment. a** Graph showing average weekly blood glucose (BG) measurements in wild-type mice treated with low-dose STZ. (green line, BG < 300, $n = 5$), (red line, BG > 300, $n = 5$), and (blue line, FAKi treated BG > 300, $n = 10$). **b** Graph showing average weekly blood glucose measurements in ElaCreERT2;R26$^{Tom}$ mice treated with low dose STZ, which reached 300 mg/dl ($n = 4$) showing that mice that received vehicle ($n = 2$) did not recover (red lines). Among FAKi-treated mice ($n = 2$), one was euthanized a week into treatment (green line, corresponding to **c**), whereas the other mouse showed improved BG values

and was euthanized 4 weeks post-FAKi treatment (blue line, corresponding to **d**). **c, d** Fluorescent imaging of Tomato in conjunction with insulin and amylase on tissues obtained from ElaCreERT2,R26$^{Tom}$ mice treated with low dose STZ ($n = 4$) followed by FAKi or vehicle ($n = 2$ per group). Mice were euthanized either 1 week into FAKi treatment (**c**), ($n = 1$), or 4 weeks after FAKi treatment cessation (**d**), ($n = 1$). Random blood glucose was checked daily at the same time to calculate the average weekly BG in (**a, b**). Arrowheads in (**c**) and arrows in (**d**) show Tom$^+$/Ins$^-$ or Tom$^+$/Ins$^+$ cells, respectively. Data in (**a**) is presented as mean ± SD. Scale bar = 20 μm. Source data are provided as a Source data file.

To study the effect of FAKi treatment following near total ablation of β-cells, 10-week-old CD1 female mice were treated with high dose STZ (150 mg/kg), or saline. Once the STZ-treated mice developed hyperglycemia (BG > 300 mg/dl), insulin pellets were implanted subcutaneously as a bridge to maintain a reasonable BG in these mice. Then, the mice from both groups were subjected to either FAKi treatment or vehicle for 3 weeks (Fig. 4a).

To evaluate the effect of FAKi treatment on glucose homeostasis, the weight of the mice was monitored on a weekly basis (Fig. 2b). In addition, daily random blood glucose measurements were performed after the initiation of the FAKi/vehicle treatment (Fig. 4c). The insulin pellets started to dissolve 5 weeks after implantation (Fig. 4c). All cohorts were subjected to an intraperitoneal glucose tolerance test (IPGTT) 4 weeks after the cessation of the FAKi treatment (7 weeks post-STZ). Consistent with our previous results (Fig. 1), FAKi treatment in non-diabetic mice did not have any substantial effects on the blood glucose, GTT or β-cell mass (Fig. 4d–g). The STZ+FAKi-treated mice showed lower BG compared to the STZ+vehicle-treated mice at 6 weeks but had higher BG than the saline+vehicle-treated control cohort (Fig. 4c). Furthermore, the STZ+FAKi-treated mice showed

partially improved glucose tolerance compared with the STZ+vehicle cohort (Fig. 4d, e). Accordingly, immunofluorescence analysis showed that the STZ+FAKi-treated animals regained approximately 30% of their original β-cell mass compared to the saline+vehicle or saline+FAKi cohorts (Fig. 4f, g). The ki67 staining revealed that the few proliferating insulin$^+$ cells (Supplementary Fig. 5) could not account for the observed increased β-cell mass in the STZ+FAKi-treated mice. To determine whether the partial β-cell mass recovery was due to higher proliferation rate among ADIP cells during FAKi treatment, diabetic Ela$^{Tom}$ mice (high dose STZ) were then treated with FAKi or vehicle for 3 weeks, during which they were given BrdU via drinking water. The mice were sacrificed immediately after cessation of the FAKi treatment. The subsequent BrdU analyses showed no difference in proliferation between the two cohorts (Fig. 4h, i).

These data collectively indicate that FAKi treatment could partially restore β-cell mass and reverse hyperglycemia in mice treated with high dose STZ. This partial recovery does not rely on proliferation of ADIP cells but is rather the result of higher conversion rate of acinar cells into insulin-producing cells. Furthermore, it shows that ADIP cells are attracted to the islets even in the absence of β-cells.

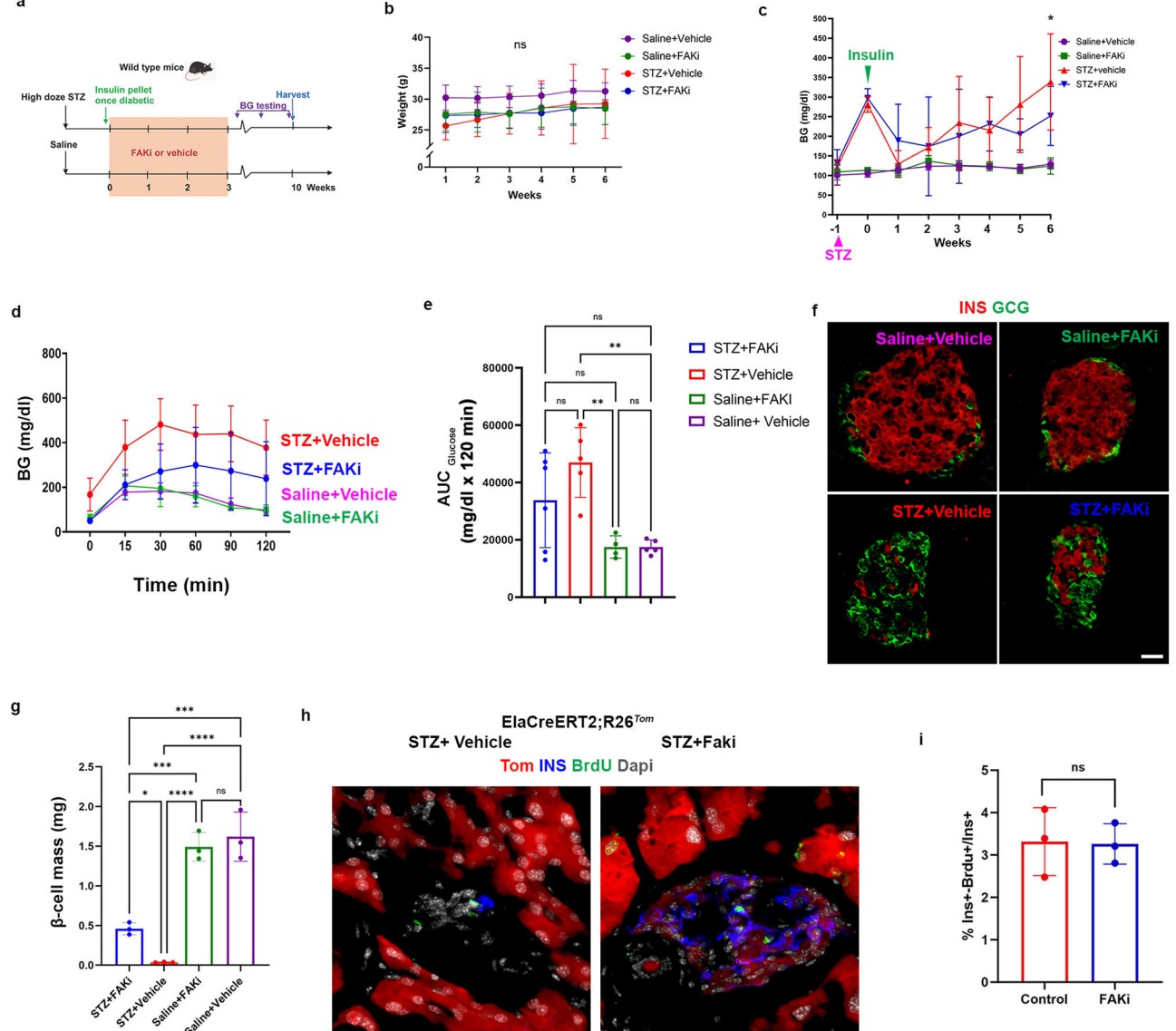

**Fig. 4 | FAKi treatment leads to partial improved blood glucose homeostasis and increased β-cells.** **a** A schematic representation of high-dose STZ-induced diabetes and FAKi treatment in wild-type mice. **b, c** Weekly measurements of the body weight (**b**) or the average random BG (**c**) in the 4 cohorts, $*p \leq 0.05$ refers to multiple comparison between STZ+vehicle or STZ+FAKi treated mice. Random blood glucose was checked daily at the same time to calculate the average weekly BG in (**c**). The timepoints for STZ treatment or insulin pellet insertion are indicated by pink (STZ) or green (insulin) arrowheads in (**c**). **d, e** Graph for glucose tolerance test (**d**) and area under the curve (**e**) performed on the four indicated cohorts ($n = 6$ for STZ+FAKi, 5 for STZ +Vehicle, 4 for Saline+FAKi, and 5 for Saline+Vehicle group). **f** Immunofluorescent

imaging for detection of insulin and glucagon of representative islets obtained from the indicated cohorts ($n = 3$ per group). **g** β-cell mass quantification in the indicated cohorts ($n = 3$ per group). **h** Immunofluorescent imaging for detection of Tomato, insulin and BrdU in STZ+vehicle or STZ+FAKi ElaCreERT2,R26$^{Tom}$ treated mice showed no differences in proliferation between the cohorts ($n = 3$ per group). **i** Quantification of ins$^+$/BrdU$^+$ cells in ElaCreERT2,R26$^{Tom}$ mice treated with STZ +vehicle or STZ+FAKi ($n = 3$ per group). Data is presented as mean ± SD. Statistical analysis was performed using two-way ANOVA (**b, c**), one-way ANOVA in (**e, g**) followed by Holm-Sidak for multiple comparisons, and unpaired two-tailed t-test (**i**). $*p \leq 0.05$, $***p \leq 0.001$, $****p \leq 0.0001$. Scale bar = 20 μm. Source data are provided as a Source data file.

## Pharmacological inhibition of FAK in a diabetic non-human primate (NHP)

The effect of FAKi on mouse acinar cells encouraged us to further investigate the potential translational benefits of FAKi for diabetes treatment. In an exploratory study we investigated the effect of FAKi on a STZ-induced diabetic NHP. To induce diabetes, the NHPs are injected with a single dose of 50 mg/kg STZ. Once random blood glucose reaches 300 mg/dl, the NHPs are managed with multiple daily injection insulin therapy (a basal and a bolus insulin). Administration of exogenous insulin allows STZ-treated NHPs to maintain BG mostly between 60 and 200 mg/dl. The blood glucose and insulin dose

generally take ~1 month to stabilize. Therefore, further interventions are typically planned for after this initial month.

In our study, four diabetic Cynomolgus macaques (males and females), which were part of an unrelated study conducted at our NHP facility, were used as non-FAKi treated cohort. These animals were between 4 and 6 years of age and weighed 4.5–7 kg. We monitored the daily BG and insulin intakes of these four diabetic animals for up to 3 months after STZ treatment. During this period, the exogenous insulin requirements of these diabetic NHPs ranged between 5 and 20 units per day (Fig. 5a, red line and Supplementary Fig. 6a). Next, we studied the FAK inhibitor in an experimental animal. To do so, FAKi

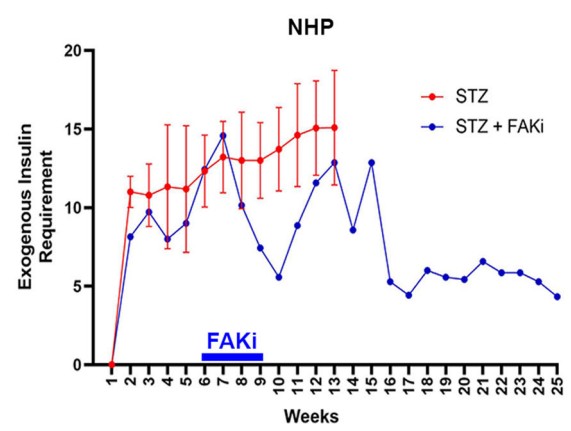

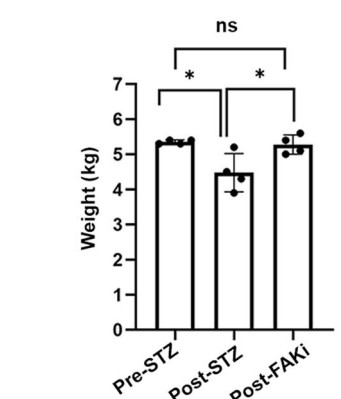

INS GCG Dapi

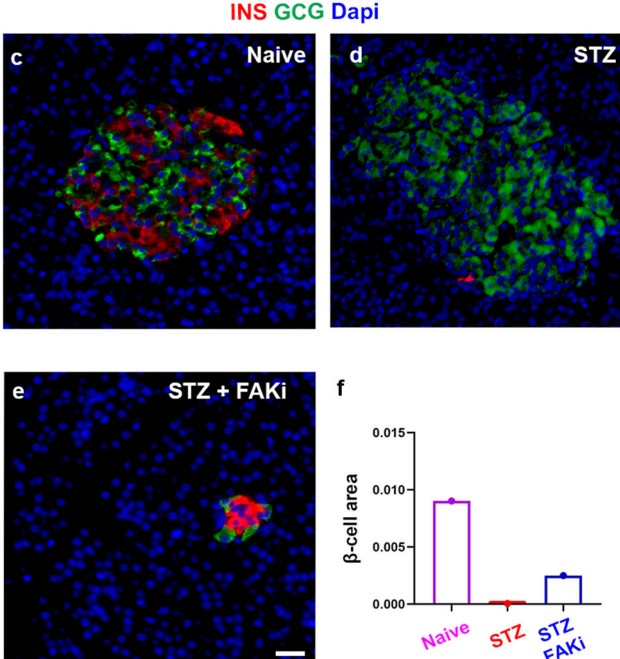

**Fig. 5 | FAKi treatment ameliorates blood glucose homeostasis in a diabetic NHP. a** Graph showing the average total weekly exogenous insulin requirements based on daily measurements in NHPs treated with STZ (*n* = 4) or STZ followed by FAKi treatment (*n* = 1). The blue bar on the x-axis denotes the period of FAKi treatment. **b** Graph showing the average weight of the FAKi-treated diabetic NHP (*n* = 1), at different occasions during indicated timepoints, including Pre-STZ treatment, Post-STZ treatment, and Post-FAKi treatment. **c**–**e** Immunofluorescent imaging for detection of insulin and glucagon of representative islets obtained from the indicated cohorts (*n* = 1 per group). **f** β-cell area quantification in the indicated cohorts. *n* = 1 NHP in each cohort. Data is presented as mean ± SD. Statistical analysis was performed using one-way ANOVA followed by Holm-Sidak test for multiple comparisons in (**b**). Scale bar = 20 μm. Source data are provided as a Source data file.

observed (Fig. 5a and Supplementary Fig. 6b). The reduced insulin requirement in the FAKi-treated diabetic NHP sustained without additional treatments until the animal was euthanized 4 months later (Fig. 5a and Supplementary Fig. 6b). Of note, the NHP maintained the same weight throughout the study (Fig. 5b). In the normal pancreas, islets composed of α- and β-cells were easily detected (Fig. 5c and Supplementary Fig. 7a). STZ treatment led to near complete ablation of β-cells (Fig. 5d and Supplementary Fig. 7b), However, in accordance with the reduced exogenous insulin requirements, we could find clusters of β-cells in the diabetic NHP that had been treated with FAKi (Fig. 5e and Supplementary Fig. 7c–e). Quantification of β-cell area from the head, the body and the tail of the pancreas revealed a significant increase in the β-cell area in the STZ+FAKi-treated NHP compared to the diabetic control NHP (Fig. 5f).

Unlike rodents, α- and β-cells are found intermixed in the human and NHP islets. However, endocrine clusters in both developing and postnatal human pancreas (up to 6 months of age) display an architecture commonly observed in the rodent islets[34]. Notably, in the STZ +FAKi-treated NHP pancreas we could find clusters of insulin-expressing cells that were surrounded by α-cells, resembling the rodent islets (Fig. 5e). This observation would further support the concept that these are newly formed islet-like structures. Together, these findings warrant more rigorous and in-depth studies to determine whether FAKi treatment would lead to β-cell neogenesis in NHPs.

## Transcriptomic analysis of FAKi-treated pancreatic endocrine islets

Our lineage tracing studies establish a progenitor-progeny relationship between acinar cells and the ADIP cells. To better characterize these newly formed insulin-producing cells, we conducted single-cell RNA sequencing (scRNA-seq) analysis on islets isolated from wild-type mice 10 days after cessation of vehicle- or FAKi treatment. Approximately 8500 cells were sequenced at 225,000 reads per cell. Only single cells with mitochondrial RNA less than 8% of the total RNA were included. Using previously established scRNA-seq data analysis pipeline (see "Methods"), a uniform manifold approximation and projection (UMAP) plot of the control and FAKi-treated cells was generated (Supplementary Fig. 8a). The overall identity of the cells was confirmed by the expression of known endocrine-, ductal-, acinar-, endothelial-, or immune cells markers (Supplementary Fig. 8b).

In our transcriptomic analysis, we focused on acinar and β-cells, as we expected the ADIP cells to display an acinar/β-cell hybrid signature. We found two clusters of acinar cells along with two clusters of β-cells in both vehicle- and FAKi-treated samples (Fig. 6a and Supplementary Fig. 8a). Surprisingly, the distinction between these two groups of acinar cells was based primarily on differential expression of both acinar genes and β-cell markers (Fig. 6b). Comparing to acinar cells in cluster 2, cells in acinar cluster 1 displayed significant lower expression of the acinar markers such as *Ptf1a, Cpa1, Amy2a2, Prss2*, and *Klk1*, but increased expression of β-cell markers *Ins1, Ins2, Pdx1, Nkx6.1*, and *Mafa* (Fig. 6b). Immunostaining analyses further confirmed the

treatment was initiated in a diabetic Cynomolgus macaques (male), approximately 5 weeks after the STZ administration, when the BG and insulin dose were stabilized (Fig. 5a, blue line and Supplementary Fig. 6b). The exogenous insulin requirements of this diabetic animal fluctuated between 5 to 20 units per day in the weeks before and after FAKi treatment. However, approximately 6 weeks after FAKi cessation, a stable near 60% reduction of exogenous insulin requirements was

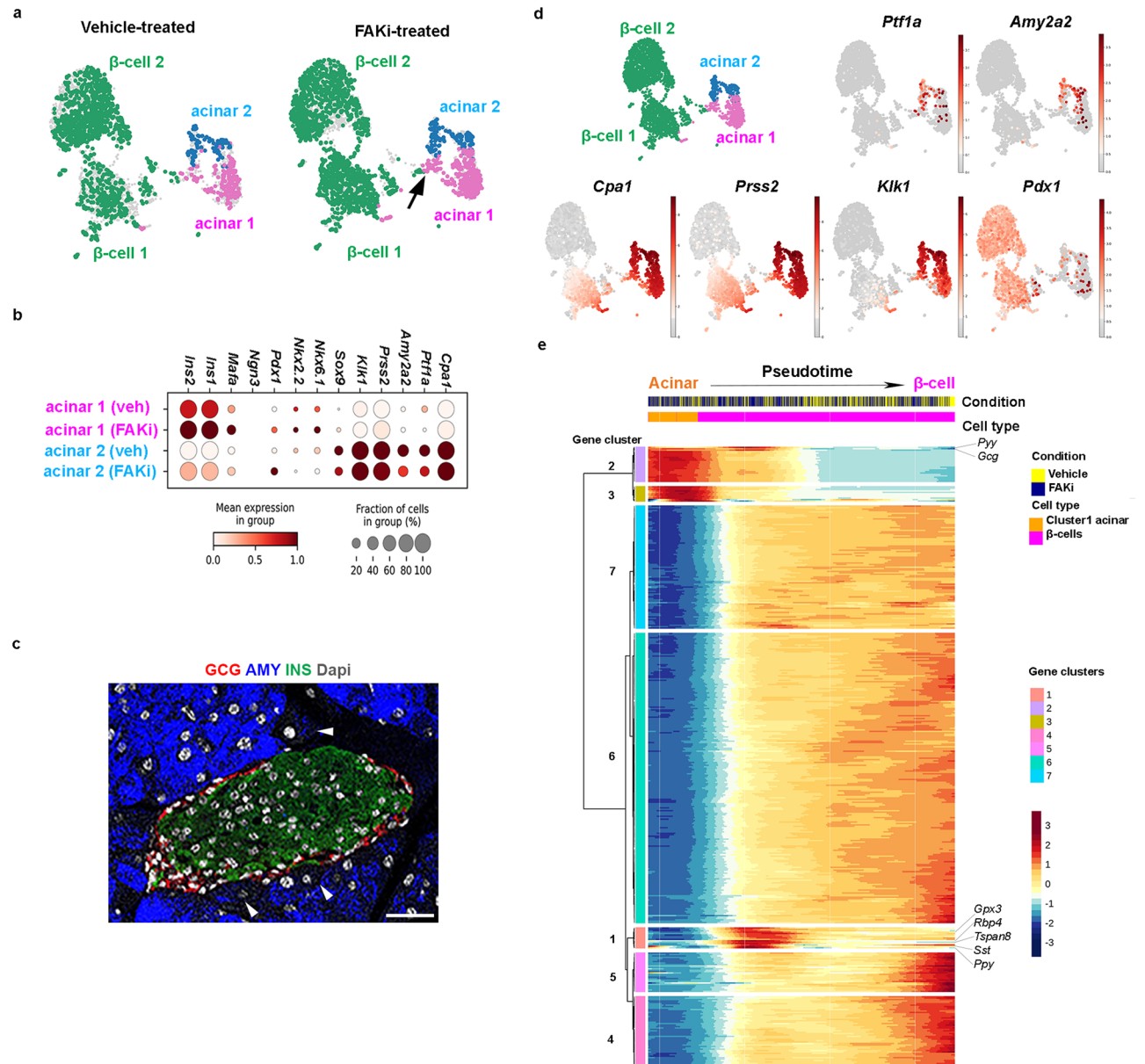

**Fig. 6 | scRNA-seq analysis of FAKi-treated acinar- and β-cells. a** UMAPs of the identified acinar or β-cell clusters defined by specific color showing vehicle- vs FAKi-treated. Arrow shows the formed bridge between acinar cluster 2 and β-cells in FAKi-treated samples. **b** Gene expression dot plots of selected marker genes across cluster 1 vs cluster 2 acinar cells in vehicle- versus FAKi-treated samples. **c** Fluorescent imaging for detection of amylase, insulin, and glucagon in wild type control pancreas. Arrowheads highlight two peri-islet acinar cells with low amylase expression. Scale bar = 20 μm, *n* = 5. **d** Single-cell gene expression of known pancreatic or acinar markers. **e** Heatmap showing the gene expression dynamics of the acinar and β-cell from acinar cells onward. Genes were grouped based on k-means clustering, and select genes are labeled on the right side of the plot.

presence of amylase[low] acinar cells in proximity of islets (Fig. 6c). Interestingly, in response to FAKi treatment, we noticed the formation of a bridge between cluster 1 acinar and cluster 1 β-cells (Fig. 6a) and a loss in *Ptf1a* expression among cluster 1 acinar cells (Fig. 6b). Notably, cluster 1 β-cell expressed higher levels of *Cpa1* and *Prss2* than cells in cluster 2 β-cells (Fig. 6d). While FAK inhibition altered gene expression patterns of cluster 1 acinar cells, it did not significantly affect expression of genes such as *Pdx1*, or *Ptf1a* expression in cluster 2 acinar cells (Fig. 6b).

Gene ontology terms (GO-term) analysis showed differential activation of insulin resistance, MODY and prolactin signaling pathways in cluster 1 acinar cells in vehicle-treated samples (Supplementary Fig. 8c). Accordingly, Differentially Expressed Genes (DEGs) analysis identified several genes with β-cell specific expression in vehicle-

treated cluster 1 acinar cells such as *Ins1*, *Ins2*, islet amyloid polypeptide (*Iaap*), glucose-6-phosphatese c2 (*G6pc2*), prolactin receptor (*Prlr*) and cyklinD2 (*Ccnd2*) as the top DEGs in cluster 1 acinar cells. G6PC2 converts glucose-6-phosphate to glucose and is predominantly expressed in β-cells[35]. IAAP is expressed in β-cells and is co-secreted with insulin in response to glucose[36], whereas PRLR mediates proliferation during pregnancy in rodents[37]. These data indicate that at baseline, acinar cells within cluster 1 not only express insulin, but also genes involved in different aspects of β-cell function. In addition, in vehicle-treated mice, GO-term of DEGs also revealed suppression of genes associated with the acinar lineage including amylase (*Amy2a2*), *Cpa1*, and elastatse (*Cel1*) (Supplementary Fig. 8c). In FAKi-treated mice, cluster 1 cells displayed further activation of insulin resistance, and FoxO signaling pathways FAKi-treated mice (Supplementary

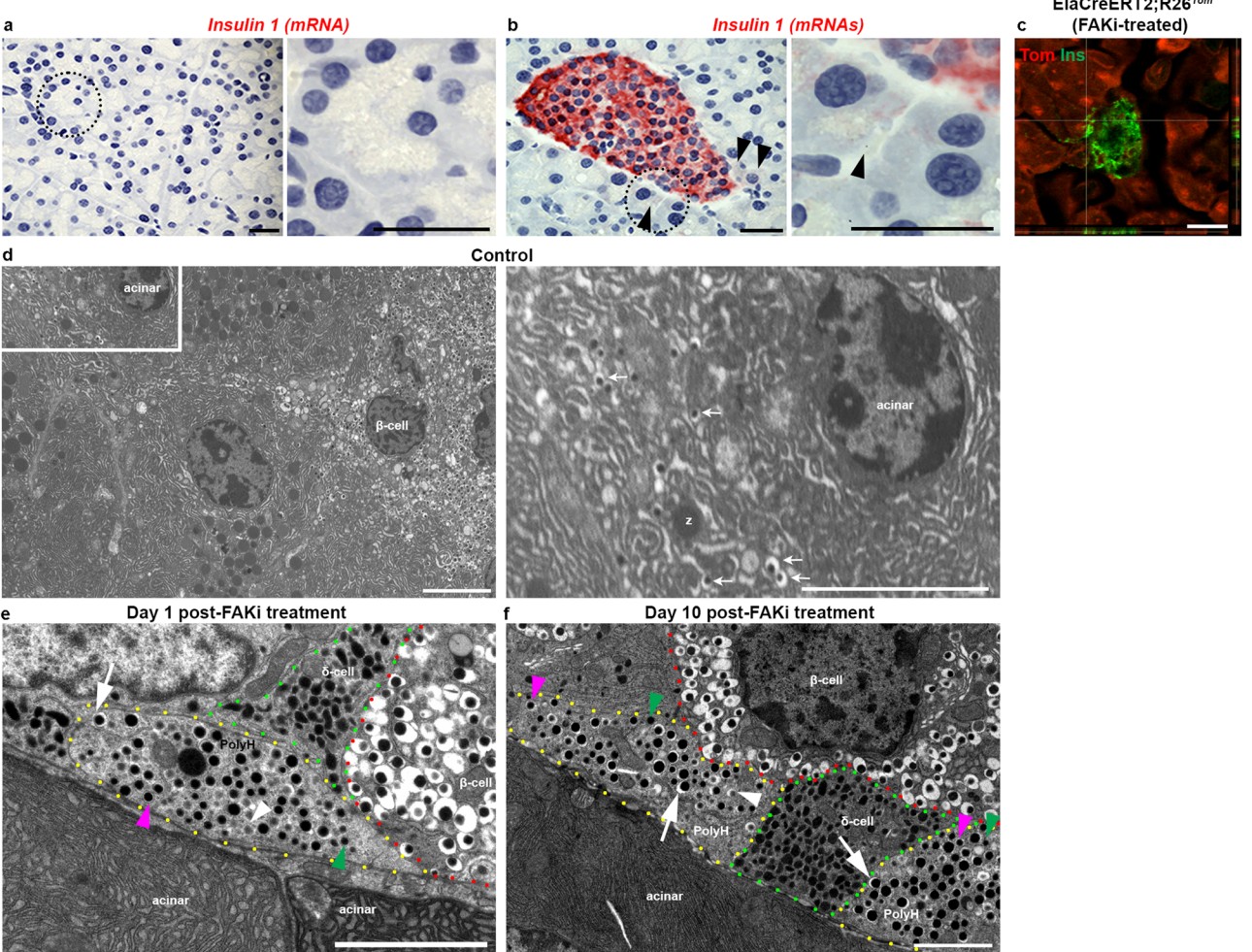

**Fig. 7 | Imaging analysis of peri-islet acinar cells. a, b** Single-molecule in situ hybridization for detection of mouse *insulin 1* in tele-islet acinar cells (**a**) or within islets and the peri-islet acinar cells (**b**) showing low levels of *insulin 1* RNA in some peri-islet acinar cells. Right panels in (**a, b**) are higher magnifications of areas marked by dotted circles in the left panels. Arrowheads in (**b**) highlight peri-islet acinar cells with low *insulin 1* expression. Scale bar = 20 µm, *n* = 3. **c** Confocal fluorescent Imaging of tamoxifen-induced ElaCreERT2,R26$^{Tom}$ mice treated with FAKi showing Tom$^+$/insulin$^+$ cells within the endocrine islets as well as in a peri-islet acinar cell. Scale bar in (**a–c**) = 20 µm. *n* = 5 mice from 3 independent experiments. **d** Transmission electron microscopy (TEM) of control mouse pancreas showing peri-islet acinar cells with β-granules (*n* = 5). Right panel in shows the Inset from (**d**) illustrating a peri-islet cell containing zymogen granules (z), and insulin granules (arrows). **e, f** TEM of mouse pancreas harvested 1 day (**e**) or 10 days (**f**) after completion of FAKi treatment showing a polyhormonal cell (outlined by yellow dotted line) that contains mature β-granule (white arrow), immature β-granule (white arrowhead), α-granule (pink arrowhead), and δ-granule (green arrowhead) (*n* = 3 per group). Scale bar = 6 µm.

Fig. 8d). Interestingly, GO-term of DEGs showed suppression of additional acinar pathways in cluster 1 with FAKi treatment such as lipase-, hydrolase-, or peptidase activity, and zymogen granule or zymogen granule membrane (Supplementary Fig. 8d). These results suggest that FAKi treatment suppresses the acinar gene signature of cluster 1 acinar cells and enhances their endocrine gene expression.

In order to reconstruct the sequence of gene expression profiles and the evolution of the ADIP cells, we performed an in-silico reconstruction of the cell lineages using slingshot and ordered them along a pseudotime (Fig. 6e). We obtained two cell lineages (acinar and β-cells), and we compared how the lineages acquire their identity by selecting differentially expressed genes (q-values < 1e10$^{-5}$) along the predicted pseudotime. We found seven gene clusters that were upregulated along different segments of the pseudotime trajectory and conducted pathway analysis to identify pathways enriched at each stage of pseudotime (Fig. 6e). Genes upregulated at the beginning of the pseudotime in gene clusters 2 and 3, included those expressed specifically in acinar cells, such as *Cela1*, *Cpa1*, *Reg1*, *Prss2*, and *Try4*. Conversely, gene clusters 4, 5, 6, and 7, were downregulated at the

beginning, but were highly expressed towards the end of the pseudotime. These four gene clusters (clusters 4–7) contained genes that can be linked to mature β-cell function[38], such as *Isl1*, *Prlr*, *NeuroD*, urocortin 3 (*Ucn3*), chromogranin B (*Chgb*), *G6pc2*, and neuropeptide y (*Npy*) (Fig. 6e). To our surprise, gene cluster 2, which otherwise showed a strong acinar signature, also included two endocrine hormones *Gcg* and peptide YY (*Pyy*). Unlike the acinar genes, these hormones were not expressed at the beginning of the pseudotime but instead only during the transition phase between acinar and β-cells (Fig. 6e). Perhaps the most intriguing cluster was the gene cluster 1, which was absent in the acinar cells but displayed its highest expression during the acinar to β-cell transition phase (Fig. 6e). Gene cluster 1 included genes associated with immature β-cells, or β-cells with reduced function such as glutathione peroxidase 3 (*Gpx3*), and retinol binding protein 4 (*Rbp4*)[38,39]. This transitory cluster was further marked with the endocrine hormones *Sst* and *Ppy* expression. Moreover, consistent with the polyhormonal signature of cells in transition, we found *MafB* and *Tspan8* expression, which are expressed in α- as well as immature β-cells, and Ppy-cell lineages[40], respectively. The

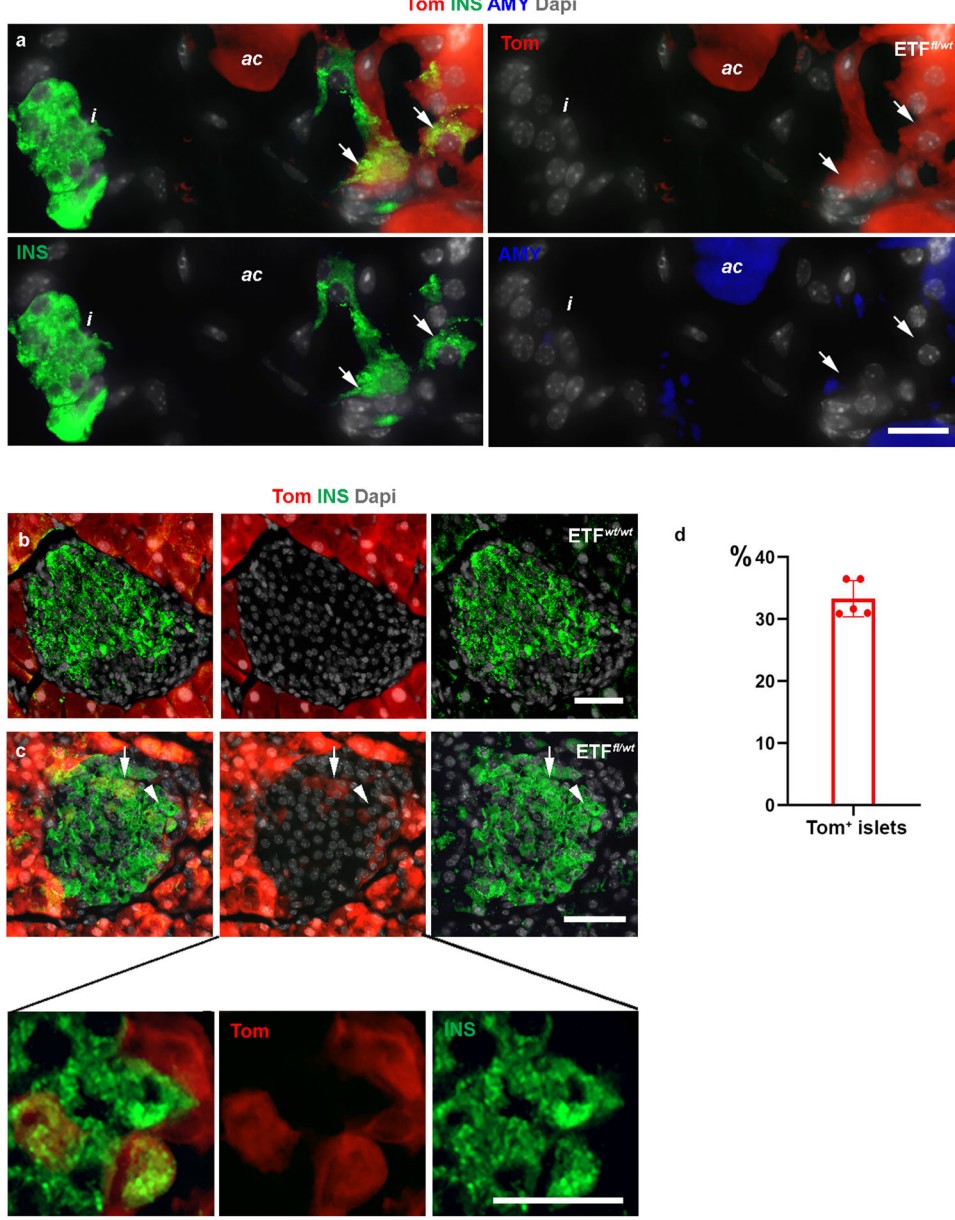

**Fig. 8 | Heterozygous deletion of Ptk2 in acinar cells leads to conversion of acinar cells into insulin-producing cells. a** Fluorescent imaging of Tomato in conjunction with insulin and amylase on tissues obtained from ElaCreERT2,R26$^{Tom}$, *Fak$^{fl/wt}$* (ETF$^{fl/wt}$) mice 1 month post tamoxifen treatment showing Tom⁺/insulin⁺ cells outside the endocrine islets (arrows). **b, c** Fluorescent imaging of Tomato in conjunction with insulin and amylase on tissues obtained from ElaCreERT2,R26$^{Tom}$, *Fak$^{wt/wt}$* (ETF$^{wt/wt}$) (**b**) or ETF$^{fl/wt}$ (**c**) mice 1 month post tamoxifen treatment showing Tom⁺/insulin⁺ cells within the endocrine islets of ETF$^{fl/wt}$ mice. **d** Quantification of islets containing Tom⁺ cells in the ETF$^{fl/wt}$ mice. Arrows in (**a**) show some Tom⁺/Ins⁺ cells outside the islets. Arrows in (**c**) show Tom⁺/Ins⁺ cells within the islets. The bottom panel in (**c**) is a higher magnification of the cells marked by arrowhead in (**c**). *n* = 5 mice from 3 independent experiments. Data in (**d**) is presented as mean ± SD. Scale bar = 20 μm. Source data are provided as a Source data file.

transcriptomic analysis show that in response to FAKi treatment, a subset of acinar cells (cluster 1 acinar cells) can convert into insulin-producing β-like cells. Furthermore, this transformation seems to entail a polyhormonal stage, where cells express glucagon, somatostatin, and pancreatic polypeptide.

### Peri-islet acinar/β-hybrid cells are a putative source for ADIP cells

Identification of a subset of acinar cells with a gene expression profile typically associated with endocrine cells was an unexpected finding and required further validation. Since the islet isolation protocol used in our studies involves handpicking of the islets, we reasoned that the two acinar clusters described in Fig. 6 are likely peri-islet acinar cells

which have been picked up together with islets during the isolation process. Therefore, to support the transcriptomic analysis, we looked for potential acinar/β-hybrid cells specifically among acinar cells surrounding the islets. First, we used the RNAscope technique to detect insulin transcripts in peri-islet acinar cells in wild type control mice (Fig. 7a, b). While the acinar cells in general were devoid of any detectable insulin transcripts (Fig. 7a), we could detect insulin mRNA in few peri-islet acinar cells (Fig. 7b). Next, we performed confocal microscopy for detection of insulin protein on pancreatic tissues obtained from FAKi-treated Ela$^{Tom}$ mice and were able to find tomato-red cells with low insulin levels next to an islet (Fig. 7c). Notably, Tom⁺/Ins⁺ cells could not be detected in vehicle-treated Ela$^{Tom}$ mice (Fig. 1k). Finally, we performed transmission electron microscopy (TEM)

(Fig. 7d–f and Supplementary Fig. 9). β-cells are characterized on electron micrographs by mature β-granules, which are secretory granules with dense cores surrounded by wide non-stained halos[41]. In addition to these more abundant mature β-granules, β-cells also contain relatively few newly formed and immature granules that display cores with much thinner halos, or no halos at all[41]. In accordance with our transcriptomic data, where a cluster of acinar cells expressed both acinar and endocrine genes, we could find mature β-granules in a subset of peri-islet acinar cells in the mouse pancreas (Fig. 7d). We next performed TEM on the pancreas of FAKi-treated mice harvested either immediately (Fig. 7e) or 10 days after cessation of the treatment (Fig. 7f). Of note, we could no longer find peri-islet acinar cells that contained β-granules in the FAKi-treated cohorts (Supplementary Fig. 9). Instead, we were able to detect cells with β-granules (proportionally fewer mature β-granules than normal β-cells) in the periphery of islets, which appeared polyhormonal (Fig. 7e, f).

Together, our transcriptomic and the subsequent validation analysis reinforce that FAKi treatment leads to conversion of acinar cells into β-like cells. Furthermore, it suggests that the peri-islet acinar cells are likely the primary cell of origin for this process.

### Heterozygous deletion of Ptk2 in acinar cells leads to conversion of acinar cells into insulin-producing cells

We have previously reported that heterozygous deletion of *Ptk2* in the developing pancreas using PdxCre results in the appearance of insulin⁺/amylase⁺ cells in the postnatal pancreas[26]. Given the observed effect of chemical inhibition of FAK on acinar cells, we next conditionally knocked out the gene encoding FAK in the acinar cells by generating ElaCreERT2;R26$^{Tom}$;*Fak$^{fl/wt}$* (ETF$^{fl/wt}$) mice. Tamoxifen was administered to 12-week-old ElaCreERT2;R26$^{Tom}$;*Fak$^{wt/wt}$* (ETF$^{wt/wt}$), and ETF$^{fl/wt}$, mice via gavage feeding, and mice were sacrificed 1 or 3 months post-tamoxifen treatment. Successful Cre-recombinase activity was evident by the expression of tomato-red in the vast majority of acinar cells. Interestingly, 1 month after induction of Cre-recombinase activity we were able to find acinar-derived insulin-producing Tom⁺/Ins⁺ cells either as single cells or in small clusters scattered within the ETF$^{fl/wt}$ parenchyma (Fig. 8a). In addition, we could detect Tom⁺/Ins⁺ cells invading 33.2% ± 2.9 of the pre-existing islets in ETF$^{fl/wt}$ mice (Fig. 8b–d). This finding indicates that deletion of one allele of *Ptk2* gene can transform a subset of acinar cells into insulin-producing cells. Our findings in the ETF$^{fl/wt}$ mice further supports that the reduction in FAK activity could trigger acinar to β-like cell conversion.

## Discussion

Diabetes mellitus is defined by high blood glucose levels caused by either reduction in the number of insulin-producing cells (T1D), or an inability of our cells to respond adequately to insulin in combination with decreased numbers of insulin-producing cells (T2D). One potential treatment would be the direct conversion of pancreatic acinar cells into β-cells in sufficient numbers to restore and maintain normal concentrations of glucose in the blood. Previous studies have shown that transduction of mouse acinar cells in vivo with vectors encoding *Ngn3*, *Pdx1* and *MafA*, three transcription factors that are necessary for β-cell development can induce conversion of acinar cells into insulin-producing β-like cells[17–19]. In this study, we show that pharmacological inhibition of FAK activity results in conversion of pancreatic acinar cells into β-like ADIP cells. Lineage tracing data showed that some ADIP cells could form larger clusters outside the islets. Given that these clusters consisted entirely of Tom⁺/insulin⁺ cells, it is unlikely that they would be part of a pre-existing islet.

Nevertheless, the ADIP cells could penetrate and invade the endocrine islets. This infiltration is significant because the islet microenvironment provides proximity to the islet blood vessels and nerves that are crucial for β-cells to function properly. Accordingly,

FAKi treatment could ameliorate the blood glucose levels in diabetic mice.

Although β-cell mass was not completely restored to normal levels in FAKi-treated diabetic mice, it resulted in nearly 30% recovery of β-cell mass compared to contribution to less than 2% of total β-cells in non-diabetic control mice. β-cells display an insignificant proliferation rate under normal conditions, whereas they tend to expand more robustly in the background of β-cell ablation[32,42–44]. Thus, the significantly higher contribution of ADIP cells to the β-cell mass in the diabetic mice compared to the FAKi-treated non-diabetic cohorts could be the result of compensatory proliferation. However, BrdU analysis ruled out proliferation of ADIP cells as the primary mechanism for the observed increase in β-cell mass in FAKi-treated diabetic mice. This could be due to the timing of FAKi treatment after STZ-mediated β-cell ablation, and that by the time ADIP cells are formed the endogenous signals that would promote proliferation (likely from macrophages) are reduced or gone. Since, the increased β-cell mass does not rely on proliferation, other factors such as higher conversion rate of acinar cells to ADIP cells due to hyperglycemia during earlier stages of FAKi treatment and/or better granulation because of improved blood glucose levels post-FAKi treatment could be the potential mechanisms for enhanced β-cell mass recovery. Nevertheless, given that in T1D patients, overt diabetes is typically manifested when 80% of β-cell mass is lost, restoring endogenous β-cell mass above this critical threshold is a significant observation in our study.

Mounting data has revealed the degree of heterogeneity that exists within acinar cells in both the adult mouse and human pancreas[45–49]. In a recent single-nucleus RNA-sequencing study, a distinct population among acinar cells in the human pancreas showed a decreased activation of acinar cell gene regulatory networks together with the capacity to potentially convert into ductal and endocrine cells[50]. Likewise, in our transcriptomic studies we identified a subpopulation of peri-islet acinar cells that displayed a decreased acinar gene expression profile in conjunction with an endocrine signature. While the existence of mouse acinar cells containing β-granules adjacent to islets is not new and has been known since the late 1960s[51], it further confirms the presented scRNA sequencing and TEM data. Our finding that these hybrid cells normally display less acinar signatures is supported by more recent reports showing that peri-islet acinar cells in the human pancreas have lower expression of acinar enzymes such as amylase and carboxyl ester lipase[52,53]. Given the heterogeneity of acinar cells, in the context of FAKi treatment, whether all acinar cells equally or a specific population can convert into β-like cells remains unknown. The inability to address this question stems from the inherent limitation of currently available acinar-cell specific promoters (including the elastase promoter used in our study) to distinguish between different existing acinar subpopulations. Nevertheless, the pseudotime analysis reported here not only confirmed our lineage tracing studies, but along with TEM data identified the peri-islet acinar/β hybrid cells as the likely source for ADIP cells. Moreover, this analysis uncovered a stepwise process in which these peri-islet cells initially lose their acinar identity, followed by a transition through a polyhormonal phase before converting into insulin-producing cells.

β-cells can take up factors released by the acinar cells as part of a crosstalk between exocrine and endocrine cells[53]. While the presence of zymogen granules in some β-cells following FAKi treatment could be due to internalization of acinar-derived proteins by β-cells, this form of crosstalk would not explain the observed changes at the transcriptional levels. The question remains though, what is the mechanism for this lineage transformation. Previous studies by other investigators have shown that ectopic expression of *Pdx1* and *Mafa* is sufficient to reprogram acinar cells to β-like cells in vivo[17]. Notably, these two transcription factors along with insulin and other genes linked to β-cell function are expressed in the peri-islet cluster 1 acinar cells. Furthermore, we found that an overall suppression of the acinar signature in

general, and *Ptf1a* in particular, was more noticeable than an increased endocrine gene expression in FAKi-treated mice, suggesting that FAKi treatment results primarily in loss of acinar gene expression in cluster 1 cells, hence allowing the existing β-cell machinery to be more pronounced. Future studies will reveal the exact mechanism by which inhibition of focal adhesion kinase activity leads to ADIP formation.

A limitation with the current study is that only one diabetic NHP was included in the FAKi-treated cohort, which does not allow to conclude that FAKi treatment would lead to improved glucose metabolism. Nevertheless, the promising outcome of the current investigation should warrant for future studies involving larger cohorts of animals and more extensive analysis such as scRNA sequencing, c-peptide and HbA1c measurements or GTT. Together, this analysis would allow us to establish whether FAKi treatment converts acinar cells into ADIP cells in NHPs.

Although, the underlying mechanism is unknown, the pharmacological inhibition of FAK for generation of insulin-producing cells is an appealing approach. Acinar cells are the most abundant cell type in the pancreas and thus can serve as a good source for potential β-cell replacement therapies without the need for immune suppression. Finally, the FAK inhibitor used in this study has passed the pharmacodynamic phase I dose-escalation trial in advanced solid tumors[29], making this compound a viable candidate for future trials for diabetes treatment in humans.

## Methods

### Study approval

All animals used in these studies were maintained according to protocols approved by the Institutional Animal Care and Use Committee of University of Pittsburgh, Pittsburgh, PA, USA (Protocol number 24024266 for mice and Protocol number 23042777 for non-human primates).

### Mice

The Rosa26$^{CAGTomato}$ (Gt(Rosa)26Sor$^{tm14(CAG-td-Tomato)Hze}$)[54] and wild-type C57bl/6 mice were purchased from The Jackson Laboratory. The *Fak$^{fl/wt}$* (B6;129X1-*Ptk2$^{tm1Lfr}$*/Mmucd)[26,55] mice were obtained from the Mutant Mouse Resource and Research Centers (MMRRC). Wild-type CD-1 mice were purchased from the Charles River Laboratories. The ElaCreERT2 strain was generated in Dr. Craig Logsdon's laboratory[7,31,32]. Both male and female mice were used for the experiments, unless otherwise stated. All mice used in this study were between 6 and 12 weeks of age at the beginning of the experiments. Mice were housed 4–5/cage and maintained in an environment at 20–23 °C and 30–70% humidity with ad libitum access to regular chow (Prolab® IsoPro® RMH 3000, LabDiet, 30005737-220) and water under a 12-h light/12-h dark cycle with the lights on from 7 am until 7 pm. Carbon dioxide inhalation was used for euthanasia, consistent with the recommendations of the Panel of Euthanasia of the American Veterinary Medical Association. Briefly, Mice were euthanized by first subjecting them to carbon-dioxide narcosis, by induction of 100% CO2 at a fill rate of 30–70% chamber volume per minute. To ensure death in an animal following $CO_2$ exposure, decapitation, or major organ harvest was done while the animal was under $CO_2$ narcosis.

### Genotyping

Genotyping was performed using KAPA Express Extract Kit (Fischer Scientific, 50-196-5275) and GoTaq® Master Mix (Promega, M7123).

Primers per mouse line were used as follows: ElaCreERT2: Forward: 5′-GCC TGC ATT ACC GGT CGA-3′, Reverse: 5′-TAT CCT GGC AGC GAT CGC-3′; R26-TdTomato: Wild Type Forward: 5′-AAA GGA GCT GCA GTG GAG TA-3′, Wild Type Reverse: 5′-CCG AAA ATC TGT GGG AAG TA-3′, tdTomato Mutant Forward: 5′-CTG TTC CTG TAC GGC ATG G-3′, tdTomato Mutant Reverse: 5′-GGC ATT AAA GCA GCG TAT CC-3′;

Fak$^{floxed}$: Forward: GAA CTT GAC AGG GCT GGT CT-3′, Reverse: 5′-CTC CAG TCG TTA TGG GAA ATC T-3′.

### Non-human primates

Cynomolgus macaques from both sexes (4–6 years old, weighing 4.5–7 kg) were purchased from Alpha Genesis Inc. NHPs were quarantined for 30 days and allowed to acclimate 1–2 weeks thereafter. Animals had free access to water and a diet of biscuits, forage mix, fruits, and vegetables. Diabetes was induced in NHPs using Streptozotocin (55 mg/kg IV)[56]. After induction of diabetes in the NHPs via IV STZ administration, NHPs received insulin therapy at a dose of ~1 unit/kg/day, 80% of the insulin dose came from long-acting insulin (Insulin glargine) divided into 2 doses, and 20% came from ultrashort-acting insulin (Insulin Lispro) that was given 10 min before meals (2 meals daily) according to a sliding scale. Blood glucoses were checked before the meals at 8 AM and 5 PM, respectively. Insulin doses were titrated daily in view of the blood glucose to maintain blood glucose ~60–200 mg/dl.

Ketamine was administered to anesthetize the NHP. Once the NHP was anesthetized, Euthasol (>150 mg/kg) was then administered IV. Signs and symptoms of death which include: decreased heart rate, loose jaw tone, decreased respiratory rate, and lack of body movement were monitored. Confirmation of death was made by the observation of no heart rate, no respirations, and no jaw tone.

### Streptozotocin treatment

Mice were treated with either low-dose (40 mg/kg) on 3 consecutive days or one time high-dose (150 mg/kg) of streptozotocin (Sigma-Aldrich, S0130) to achieve partial or near-complete β-cell ablation, respectively.

### Tamoxifen treatment

For Cre-recombinase activation, ElaCreERT2;R26$^{Tom}$, ElaCreERT2;R26$^{Tom}$;*Fak$^{wt/wt}$*, and ElaCreERT2;R26$^{Tom}$;*Fak$^{wt/fl}$*, mice were gavaged once daily with 5 mg Tamoxifen (Sigma-Aldrich, T5648) dissolved in corn oil for 3 days, as described elsewhere[57,58].

### Insulin pellets implantation

LinBit implants (LinShin Canada Inc., As-10L) were placed subcutaneously under the mid dorsal skin. The dosages used in this study were according to the manufacturer's recommendation. Briefly, 2 LinBit implants were used for the first 20 g in body weight. For each additional 5 g, another implant would be added.

### FAK inhibitor treatment

PF-562271 (Medkoo Biosciences, Inc., 202228) was dissolved in 5% DMSO and administered with 5% Gelucire® 44/14 (Gattefosse) dissolved in sterile water. PF-562271 (50 mg/kg) was given via oral gavage (20 Gauge) twice a day, as described elsewhere[28].

### Immunohistology

Tissue processing, and immunostaining were performed as previously described[26,32,58–60]. Briefly, harvested pancreata were fixed overnight at 4 °C in 4% paraformaldehyde and were either processed for paraffin embedding or incubated in 30% sucrose solution overnight at 4 °C and subsequently embedded with OCT compound. Sections were permeabilized with 0.1% PBS/Triton X-100, washed in PBS and blocked for 30 min in 10% normal donkey serum (NDS) in 0.1% PBS/Triton X-100. Primary antibodies were incubated overnight at 4 °C, while secondary antibodies were incubated for one hour at room temperature.

The following antibodies were used: goat anti-Amylase (1:250, Santa Cruz, sc-12821); rat anti BrdU (1:100, Abcam, ab6325); rabbit anti-Glucagon (1:1000, Linco/Millipore, 4031-01 F); rabbit anti-Glut2 (1:50, Santa Cruz, sc-31825); guinea pig anti-Insulin (1:400, Abcam, ab195956); rabbit anti-Insulin (1:500, Abcam, ab181547); rat anti-ki-67

(1:250, Invitrogen, 14-5698-82); rabbit anti-Nkx6.1 (1:150, Abcam, ab221549); goat anti-Pdx1 (1:100, Abcam, ab47383).

All of the following secondary antibodies used for immunostaining were purchased from Jackson ImmunoResearch Laboratories: biotin-conjugated anti-rabbit (1:500, 711-066-152), biotin-conjugated anti-rat (1:500, 712-066-153), biotin-conjugated anti-guinea pig (1:500, 706-065-148), biotinconjugated anti-goat (1:250, 705-065-147); Cy2-conjugated streptavidin (1:500, 016-540-084); Cy3-conjugated streptavidin (1:500, 016-160-084); Cy5-conjugated streptavidin (1:100, 016-600-084); Cy2-conjugated anti-guinea pig (1:300, 706-545-148), Cy3-conjugated anti-guinea pig (1:300, 706-166-148), Cy2-conjugated anti-rabbit (1:300, 711-485-152), Cy3-conjugated anti-rabbit (1:300, 711-165-152), Cy2-conjugated anti-rat (1:300, 712-545-153), Cy3-conjugated anti-rat (1:300, 712-166-150), Cy2-conjugated anti-goat (1:300, 705-545-147) and Cy3-conjugated anti-goat (1:300, 705-165 147).

## Single-molecule in situ hybridization
Tissues were fixed in 4% PFA and processed for paraffin embedding, and cut into 5 μm sections. The RNAscope Red Detection kit (Advanced Cell Diagnostics, 322360-USM) was used for detection of mouse *insulin 1* (ACD, 414661) according to the manufacturer's instructions.

## Fluorescent Imaging
Imaging of pancreatic tissue sections was performed using a Leica Dmi8 fluorescent light microscope at 10×, 20×, or 63x objectives using LASX software. The microscope is equipped with 405, 488, 568, and 647 nm filters.

## Confocal imaging
Sections were imaged using a Leica Stellaris 5 confocal laser scanning microscope at 20× or 63× objectives using LASX software. Final figures were composed using Adobe Photoshop.

## Transmission electron microscopy
TEM was performed as previously described[61]. Briefly, after fixation (2.5% glutaraldehyde, 2% PFA in 0.1 M sodium phosphate buffer), tissues were rinsed twice in 0.1 M monosodium phosphate for 30 min and then placed in 1% osmium tetroxide in water for 1 h. Tissues were rinsed twice in deionized water. The samples were then dehydrated in serial concentrations of ethanol. Then tissues were placed were pre-infiltrated in half resin/half propylene oxide overnight. The next day, tissues were infiltrated in 100% resin for 5 h, and were then embedded with fresh resin and polymerized at 60 °C overnight. The embedded tissues were sectioned with a Leica EM UC6 ultramicrotome. The sections were stained with 4% aqueous uranyl acetate for 30 min and 2 min in 0.2% lead citrate in 0.1 N sodium hydroxide (NaOH). TEM imaging was performed using Thermo Scientific Ceta 16 MP camera.

## Intraperitoneal glucose tolerance test (IPGTT)
Overnight 16-h-fasted mice were injected I.P. with 2 g/kg glucose (Sigma-Aldrich, G7528). Blood glucose was measured from the tail vein at 0, 15, 30, 60, 90, and 120 min after injection using glucometer (Contour NEXT EZ)[61].

## BrdU-labeling
BrdU (0.8 g/L) (Sigma-Aldrich, B9285) was provided in drinking water during the FAKi treatment with the water changed daily.

## Measurement of serum pancreatic enzymes
Approximately 50 μl of blood from the tail vein were collected via the Microvette CB300 Capillary Blood Collection Tube. Serum pancreatic enzyme concentrations were measured using a mouse amylase ELISA kit (LifeSpan BioSciences, LS-F9289) and mouse lipase ELISA kit (Aviva System Biology, OKCD07269).

## Islet isolation
Pancreas digestion and islet isolation Islets (approximately 1200 islets for each cohort) were isolated from the vehicle- ($n = 5$) or FAKi-treated ($n = 5$) mice following a previously published protocol[62]. Briefly, the pancreatic duct was infused, and the pancreas was subsequently digested with type V collagenase (1.4 mg/ml). The islets were separated from the exocrine tissue with the Histopaque gradient solution (100 ml Histopaque 1077 and 120 ml Histopaque 1119) (Sigma-Aldrich, 10771 & 11191) and then washed with Hanks' Balanced Salt Solution (Gibco) containing 20 mM Hepes buffer (Gibco) and 0.2% bovine serum albumin (Sigma-Aldrich). The islets were then handpicked to eliminate contamination from exocrine tissue.

## 10X Genomics (single-cell RNA-Seq)
Single cells were stained using trypan blue for visualization, counting, and viability assessment using the Countess II Automated Cell Counter and diluted as needed to a final concentration of 700–1200 cells/μl in 1x PBS + 0.04% BSA. A minimum of two reproducible counts with SD < 25% were required before droplet preparation. Cell viability was 75–76%. A total of 8832 for the vehicle- and 8437 cells for the FAKi-treated samples were used for sequencing. Cells were processed with Chromium Next Gem single Cell 3′ GEM Library and Gel Bead Kit v 3.1 (10X Genomics) according to manufacturer instructions. Briefly, cells diluted in master mix containing reverse transcription reagents and primer and were transferred to the ChromiumChip with gel beads and partitioning oil for preparation of nanoliter-scale Gel Bead-In-Emulsions (GEMs). Reverse transcription produced cDNA with a cellular 10X barcode and UMI, and was recovered with Dynabeads MyOne SILANE magnetic beads. Subsequently, the cDNA underwent 12 cycles of amplification before clean-up using SPRIselect Reagent (Beckman). Total cDNA yield was calculated following quality assessment (High Sensitivity D5000 ScreenTape, Agilent) and fluorometric quantitation (Qubit, ThermoFisher).

Libraries were constructed by enzymatic fragmentation, end repair, and A-tailing according to the manufacturer's instructions. A double-sided SPRIselect Reagent (Beckman) clean up prepared samples for adapter ligation and another round of magnetic bead clean-up before sample indexing PCR for 12 cycles. A further double-sided size selection produced libraries of ~400 bp, which were quantified for paired-end sequencing (26 bp × 98 bp).

The standard 10x Genomics protocol (CellRanger V6.0172) was used to generate gene-cell expression matrices. Preliminary QC including correcting for batch effects on the FASTQ files was performed (CellRanger) before aggregating the libraries. Seurat (V3.1, R package) was used to determine filter criteria according to the median number of genes and mitochondrial gene percent via QC plots. Gene counts were normalized using SCRAN (v1.22.1)[63] for each sample separately. Log-transformed counts using natural log and pseudocount 1 were calculated for downstream integration. Highly-variable genes (HVGs) were calculated per sample[64] to select HVGs unaffected by sample variance. Samples were integrated with scanorama[65] (v1.7.1) using 8000 HVGs. Clustering was performed on the k-nearest neighbor graph (k = 15) calculated from the integrated embedding using Louvain clustering[66] with resolution 1. The integrated embedding was reduced to 50 principal components before calculating the neighborhood graph. Cell types were annotated based on canonical marker genes, and unbiased cell type annotation using relevant references. Differentially expressed genes were identified across the clusters, and false discovery rate correction was done using Benjamini–Hochberg correction. Functional enrichment was done using clusterProfiler resources from MSigDB. Pseudotime analysis was performed using Slingshot, and differentially expressed genes along the trajectory were identified using Monocle2, and visualized as heatmaps ordered based on the pseudotime.

## Measurements of β-cell mass

β-cell mass quantification was performed as previously described[67]. Briefly, sections at 200–300 μm intervals of the whole pancreas were stained with insulin and glucagon. The cross-sectional area of β-cells and cross-sectional area of total tissue were measured by ImageJ software. The β-cell mass of each pancreas was calculated by multiplying the average value of relative cross-sectional area of each cell type and the weight of the pancreas.

## Measurements of β-cell area

To estimate the β-cells area, NHP pancreas from naive, STZ-treated only or STZ+FAKi treated were harvested, sectioned, and immunostained for insulin and glucagon. Sections (8–10 μm) were collected serially so that each slide would contain semi-adjacent sections across the entire tissue. Data were obtained by analyzing different sections of pancreas i.e., from head, body, and tail. Ratio of Insulin/Pancreas area were quantified using ImageJ software.

## Quantification analysis

To calculate the percentage of intra-islet $Tom^+/Ins^-$, or $Tom^+/Ins^+$ cells in FAKi-treated mice, whole pancreata were sectioned, and sections separated by 200–300 μm were stained for insulin. To calculate the percentage of $BrdU^+/Ins^+$ cells in mice, whole pancreata were sectioned, and sections separated by 200–300 μm were stained for insulin and BrdU. Data were obtained by analyzing ~70–80 islets per mouse. Captured images of islets were analyzed using ImageJ software.

## Statistical analysis

AUC for GTT was calculated by the trapezoidal method. Comparisons between 2 groups were made using unpaired, two-tailed t-test as indicated. Comparison between multiple groups was made using one-way or two-way ANOVA as indicated followed by the Holm-Sidak test for multiple comparisons. $P < 0.05$ was considered as statistically significant. Statistical analysis was performed using GraphPad Prism 9 software (GraphPad software version 9.2.0, San Diego, CA). Unless specified, data in the text, table and figures is expressed as a mean ± standard deviation (SD).

## Reporting summary

Further information on research design is available in the Nature Portfolio Reporting Summary linked to this article.

# Data availability

The RNAseq data generated in this study have been uploaded to the Gene Expression Omnibus under accession number: GSE251852. Source data are provided with this paper.

# Code availability

All the analysis was carried out using available Bioconductor packages, and no custom scripts were generated.

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

## Acknowledgements

The authors thank Nina Romano for her assistance in electron microscopy imaging. We also acknowledge the support of the mouse facility and the histology core at the Children's Hospital of Pittsburgh Research Center. In addition, we thank Dr. Irka Redelsperger for her assistance and care of the NHPs. The Leica Stellaris 5 confocal microscope at the Rangos Research Center Cell Imaging core facility was purchased using an internal support grant from the Children's Hospital. We acknowledge the Cell imaging core staff for their support. This project used the University of Pittsburgh HSCRF Genomics Research Core, RRID: SCR_018301 single cell services and the University of Pittsburgh Health Sciences Sequencing Core at UPMC Children's Hospital of Pittsburgh, sequencing service. This research was supported in part by the University of Pittsburgh Center for Research Computing through the resources provided. Specifically, this work used the HTC cluster, which is supported by NIH award number S10OD028483. This work was supported by NIH grants R01DK101413 (to F.E.) and R01DK120698 (to G.K.G. and F.E.), R21AI158824 (to F.E.), CA236965 (to J.H. and F.E.), 1K08DK129834 (to M.S.), JDRF grant 2-SRA-2022-1211-S-B (to J.D.P. and F.E.), Research Advisory Committee (RAC) Children's Hospital of Pittsburgh of UPMC (to F.E.), Cochrane-Weber Endowed Funds for Diabetes Research (to F.E.) and The Children's Hospital of Pittsburgh of UPMC (to F.E.).

## Author contributions

S.D., U.A.R., M.S. and F.E. designed the experiments. U.A.R., J.R.A., S.D., A.H. and M.S. investigated the effect of FAKi on diabetic animals. U.A.R., J.R.A., S.D. and F.E. performed the lineage tracing studies. M.S. measured exocrine enzyme activity and performed GTT assessments. U.A.R., J.R.A. and F.E. generated and analyzed the ElaCreERT2;R26$^{Tom}$;$Fak^{fl/wt}$ mice. A.S. isolated islets from the vehicle- or FAKi-treated mice. D.R. and K.Y. performed the bioinformatic analysis. M.S. and R.K. performed the electron microscopy analysis. S.D., S.Y. and K.P. performed confocal imaging. M.S., S.D., J.H., H.L., J.D.P., G.K.G. and F.E. interpreted results. F.E. wrote the original draft. S.D., M.S., G.K.G. and F.E. reviewed and edited the manuscript.

## Competing interests

Farzad Esni and Jing Hu are listed as inventors on a U.S. Patent Application based on this work. The other authors declare no competing interests.
