## [Peer Review File · Nature Communications]

Acinar to β -like cell conversion through inhibition of focal adhesion kinaseEditorial Note: This manuscript has been previously reviewed at another journal that is not operating a transparent peer review scheme. This document only contains reviewer comments and rebuttal letters for versions considered at *Nature Communications*.

REVIEWER COMMENTS

Reviewer #1 (Remarks to the Author):

Dahiya et al present intriguing data on the conversion of pancreatic acinar cells to insulin producing cells when FAK activity is inhibited with an orally given small molecule. This is a revised version that had been submitted to a sister journal previously. This study was based on this lab's previous work showing that inhibition of FAK activity in embryonic mouse pancreas promoted endocrine differentiation and prevented acinar differentiation with FAK being required both for adult acinar integrity and maintenance. Using ElastaseCreERT;R26tomato mice for lineage tracing, they show with time after treatment Tom+Insulin + cells adjacent and within the islets of STZ diabetic mice. The Tom+Ins+ contributed 1.8% of the beta cells and were found in 40% of the islets. They further characterised the cells with sc RNA seq and ultrastructure (transmission EM). They identify a cluster of acinar cells (cluster 1) that express low levels of beta cell genes that is presumed to be the source of the Tom+Insulin + cells. With the FAKi treatment they found some amelioration of blood glucose in either low or high dose STZ diabetic mice and lower insulin requirements in 1 STZ diabetic non-human primate. They conclude that FAKi treatment suppresses the acinar signature and allows a subset of acinar cells (cluster 1 in RNAseq) that express beta cell genes at low level to develop stronger beta cell phenotype.

In response to a previous review for a sister journal, the authors have made a number of revisions but still fall short on providing all the necessary convincing data on their exciting but provocative finding. Some of the issues remaining are:

1. To convincingly show that the insulin+ cells are derived from Tomato+ acinar cells, confocal images as z stacks are needed. Since frozen sections are used, there is concern about "bleed" through of color from cells below the surface. Only Fig2D is confocal and all the rest are not. Additionally with the known robustness of the Logsdon mouse model of ElaCre ERT and its original report of even 50% of the acinar cells labelled in adults even without tamoxifen, it would also be useful to show parallel images of ELACreERT;R26tom without the FAK inhibitor as control.
2. Previously I raised the question of how can the acinar conversion (only 1.8% of the insulin + cells) lead to the increased beta cell mass after High dose STZ (30% increased) seen in Fig 4g. The authors's response was that it was "the result of proliferation". Yet the authors clearly state that none of the intraislet Tom+ cells were BrdU+ (p5, Fig 2e), thus suggesting acinar conversion not proliferation of the ADIP cells. Ext Fig 4 does show an extraislet insulin + cell expressing ki67, but to conclude that the increased beta cell mass after HD STZ was due to proliferation of single Insulin- expressing cells before they formed clusters needs supporting quantification. Additionally is the suggestion that clumps
3. The scRNAseq data (Fig 6) provides the interesting two acinar clusters. What percentage of the acinar cells are in each cluster? Does the percentage change after FAKi as might be expected if there has been the conversion to insulin producing cells? Also it would be stronger to also provide cluster 2 in the graphs in Fig6c
4. The TEM images have been greatly improved in quality. However, their interpretation seems inaccurate. The secretory granules are how one identifies islet cell types ultrastructurally. In Fig 7B and C, what are labeled as ADIPs have the characteristic granules of the glucagon-producing alpha cell and what are labelled as the alpha cells are likely delta cells. What is labelled z (zymogen granule) in the control beta cell (Fig 7b and 7c are probably autophagic vacuoles/autophagosomes?

5. The methods are skimpy at best. While it is permissible to say as "previously described", it is necessary to cite the reference. Under the TEM, what is MJK solution? Karnovsky's fixative? Again usually one references Reynold's lead citrate etc.

Specific points;

While it is understandable to have few nonhuman primates, giving 4 data points and statistics on n=1 /cohort (Figure 5f) is not appropriate.

It is difficult to understand Fig 1H. There are many Insulin + cells (many nuclei) in a space that is only 20 um, so how is this considered extraislet and not an islet?

There is no legend for Fig 3b.

Fig 4e. The AUC for IPGTT for STZ+FAKi and STZ+ Vehicle are not significantly different, so one cannot say there is improved glucose homeostasis as in figure legend title and p8 text?

Fig 4H. How is it determined that this islet is neogenic?

Ext. Figure 4. It is unclear how the 3 replicating (Ki67+) cells outside the islet are determined to be acinar cells without immunostaining?

Reviewer #2 (Remarks to the Author):

The author has, in general, been responsive to the comments made by the reviewer to the earlier version. Indisputably, the revised manuscript has more clarity, and the results are presented in a more compelling fashion. However, there are still a few loose ends that would need to be addressed. It is also customary to point the reviewer to the exact place in the revision where the changes have been made, rather than merely stating "supporting data have been added". This has not been done consistently but should be done in the context of a second revision.

1. Regarding point 5, the original question was: "The findings described in extended figure 1 are confusing --Single endocrine cells next to an islet-- Islets are heterogeneous and it is unclear the significance of this observation. Were these purported occurrences more commonly seen in FAKi-treated mice than in controls? If so, the observation has to be carefully defined (as in "presence of peri-insular endocrine cells as determined by immunostaining of insulin") and quantified vs. controls. Otherwise, these are at most anecdotal observations.

The author states in the rebuttal:

"Single cells such as one shown in Extended Figure 1, was occasionally seen only in the FAKi treated mice. We have modified the text to address this issue"

However, the text remains:

"Another intriguing feature found in the pancreas of FAKi-treated mice was the appearance of clusters of endocrine-like cells within acini (Extended Fig. 1c, c").

This still needs clarification. Since islets are naturally interspersed with acini, what makes a "cluster of endocrine-like cells" different from a small islet? Also, immunostaining for INS is still necessary. The reviewer understands that subsequent studies do show this. However, this is the first time that this is mentioned, so proof must be shown at this time, or at least the text modified accordingly.

2. Regarding the reviewer's comment that Extended figure 4 is very difficult to interpret: The

original critique was that "according to Ext. Fig. 4b, no animals became diabetic. This is in contrast with the data shown in 4c. If insulin pellets last until week 5, why does glycemia not rise above 150 mg/dl in any group after that time point? Please clarify".

The author responds that "the earliest values (week 1) refer to the first readings after insulin pellet implantation. Therefore, the BG values are below 300. As demonstrated, once the pellet started to resolve (around week 5) the STZ-vehicle cohort displayed BG values above 300. We have modified the figure and the legend to clarify." A corrected graphic is shown.

The reviewer appreciates the clarification. However, it is not appropriate to omit the readings prior to week one. The diabetic state needs to be shown. The green arrow points at some day between day 1 and 7, and by then we are told that the animals are already diabetic. The most widely accepted standard for diabetic mice is three consecutive readings above 250 mg/dl. This needs to be shown. If the x axis unit is weeks, then it would be helpful to have a side chart corresponding to the interventions and readings happening during the first seven days. A supplementary table with the data would also be acceptable.

It is also unfortunate that there is only one time point (week 6) where there is some meaningful degree of separation between the FAKi and vehicle STZ groups, and, given the error bars, it is probably not statistically significant. If the trend was towards separation, the authors must discuss why the experiment did not go on for longer, and at any rate state the p values corresponding to the statistical differences between groups.

3. Regarding 12, the original critique was: These data indicate that like b-cells, ADIP cells display an insignificant proliferation rate under normal conditions, whereas they tend to expand more robustly in the background of β -cell ablation". Where are the BrdU data to substantiate this claim?

The authors respond: "The increased beta-cell mass is the result of proliferation (supporting data has been added). However, other factors such as higher conversion rate of acinar cells as due to hyperglycemia during earlier stages or better granulation because of improved blood glucose levels cannot be ruled out. We mention these issues in the Discussion."

In relation to this, the supporting data added by the authors are in Extended figure 4, where a single INS+/Ki67+ cell is shown. This is in no way the data that supports the above claim. If the authors believe that their data are consistent with beta-cell proliferation (rather than acinar-to-beta cell conversion, as in non-stz-treated mice), proper quantification needs to be shown. Ki67, not being cumulative, may not be the right tool to use: BrdU treatment to the live mice during the treatment would. In summary, as this claim is not supported by the presented data, it should be either removed or discussed as speculation.

4. Regarding original critique 25: "These cells were more prevalent in the pancreas collected two weeks after FAKi, treatment (Fig. 5d, d'). This needs to be quantified."

The author states "We have rephrased this sentence".

However, the rephrasing is merely something along the lines of "the cells seemed more prevalent", which, frankly, looks like a rather easy way out of the reviewer's request. Any top journal (and this one is surely no exception) is very strong on rigor and statements being thoroughly backed by hard data. The use of expressions like "seemed" when applied to actual quantifiable data is not appropriate. The authors are encouraged to offer quantification or remove this statement altogether.

Finally, it would be helpful for the reviewer to see the actual scRNAseq files, including FASTQ files and differential expression data by cluster, if available.

Reviewer #3 (Remarks to the Author):

The authors present that treatment of adult mice and NHP with a small molecule that specifically inhibits kinase activity of focal adhesion kinase results in trans-differentiation of acinar cells into insulin producing β -like cells. It would be highly significant if the drug could increase beta cell volume and improves diabetes.

However, there are several questions as to whether the data supports the conclusion.

Major comments:

1. Since pancreas transplants and islet transplants are being performed, the following statement in the abstract is inappropriate: "Insufficient functional β -cell mass causes diabetes; however, an effective cell replacement therapy for curing diabetes is currently not available".
2. I think that the authors have not adequately addressed many of the issues raised by the reviewers in the last submission. In addition, the number of individuals in animal experiments was generally small, so the basis for conclusions might be weak.
3. Why is the number of FAKi-treated NHPs one? If you are going to state scientific findings from this, please provide data from at least some individuals, I understand the difficulty of experimenting with NHPs, but I also understand the magnitude of variation on the other hand.
4. Please show the blood glucose data for the NHPs.
5. Please indicate the algorithm for determining the amount of insulin to be administered to the NHPs.
6. If ADIP cells are increased, is there a way to address type 1 diabetes autoimmunity?
7. Assuming an increase in insulin-producing cells in the animal, please indicate the blood insulin or C-peptide concentration, if possible.
8. The effect of FAKi does not seem to be as great in mice or in NHP; extending the duration of FAKi administration or increasing the dosage should be considered.
9. The authors mentioned that the new ADIP cells will infiltrate into the islets, but how will they move from their original location, which pathway will they infiltrate from, and in which space will they settle to live?
10. Is there a difference in the number and function of islets isolated from diabetic control and FAKi-treated mice?
11. If possible, please describe glucose tolerance test data and HbA1c data from the NHP experiment.

General statement

We would like to thank the reviewers for their comments. The revised manuscript is certainly an improved version of our original submission, mainly because of the feedback we received from the reviewers. Most of the comments raised by the reviewers could be addressed by limited additional experiments, better clarification, modification of the figures, or adding more experimental details, as reflected below. In addition, we provide evidence (BrdU analysis combined with lineage tracing) which rules out proliferation as the primary mechanism for the observed increase in β -cell mass in FAKi-treated diabetic mice.

Furthermore, the reviewers have been granted access to the FASTQ files, as requested.

<https://www.ncbi.nlm.nih.gov/geo/query/acc.cgi?acc=GSE251852>

Token: **olebgmuohxiplmv**

Reviewer #1 (Remarks to the Author):

1. To convincingly show that the insulin+ cells are derived from Tomato+ acinar cells, confocal images as z stacks are needed. Since frozen sections are used, there is concern about “bleed” through of color from cells below the surface. Only Fig2D is confocal and all the rest are not. Additionally with the known robustness of the Logsdon mouse model of ElaCre ERT and its original report of even 50% of the acinar cells labelled in adults even without tamoxifen, it would also be useful to show parallel images of ELACreERT;R26tom without the FAK inhibitor as control.

To address this important concern, we have added new confocal images with orthogonal views of z stacks, (Fig. 1k,l), (Fig. 2d) and (Fig. 7c). Furthermore, the new figures 1i and 1k show vehicle-treated ElaCreERT,R26Tom mice.

2. Previously I raised the question of how can the acinar conversion (only 1.8% of the insulin + cells) lead to the increased beta cell mass after High dose STZ (30% increased) seen in Fig 4g. The authors’ response was that it was “the result of proliferation”. Yet the authors clearly state that none of the intraislet Tom+ cells were BrdU+ (p5, Fig 2e), thus suggesting acinar conversion not proliferation of the ADIP cells. Ext Fig 4 does show an extraislet insulin + cell expressing ki67, but to conclude that the increased beta cell mass after HD STZ was due to proliferation of single Insulin- expressing cells before they formed clusters needs supporting quantification.

Additionally is the suggestion that clumps

This sentence was not complete, and therefore could not be addressed.

We have performed the recommended BrdU study and have modified the text accordingly:

Page 8 (Result section):

“The ki67 staining revealed that the few proliferating insulin⁺ cells (Extended Fig. 4) could not account for the observed increased β -cell mass in the STZ+FAKi-treated mice. To determine whether the partial β -cell mass recovery was due to higher proliferation rate among ADIP cells during FAKi treatment, diabetic Ela^{Tom} mice (high dose STZ) were then treated with FAKi or vehicle for three weeks, during which they were given BrdU via drinking water. The mice were sacrificed immediately after cessation of the FAKi treatment. The subsequent BrdU analyses showed no difference in proliferation between the two cohorts (Fig. 4h). These data collectively indicate that FAKi treatment could partially restore β -cell mass and reverse hyperglycemia in mice treated with high dose STZ. This partial recovery does not rely on proliferation of ADIP cells but is rather the result of higher conversion rate of acinar cells into insulin-producing cells”.

Page 15 (Discussion):

“Although β -cell mass was not completely restored to normal levels in FAKi-treated diabetic mice, it resulted in nearly 30% recovery of β -cell mass compared to contribution to less than 2% of total β -cells in non-diabetic control mice. β -cells display an insignificant proliferation rate under normal conditions, whereas they tend to expand more robustly in the background of β -cell ablation^{31,41-43}. Thus, the significant higher contribution of

ADIP cells to the β -cell mass in the diabetic mice compared to the FAKi-treated non-diabetic cohorts could be the result of compensatory proliferation. However, BrdU analysis ruled out proliferation of ADIP cells as the primary mechanism for the observed increase in β -cell mass in FAKi-treated diabetic mice. This could be due to the timing of FAKi treatment after STZ-mediated β -cell ablation and that by the time ADIP cells are formed the endogenous signals that would promote proliferation (likely from macrophages) are reduced or gone. Since, the increased β -cell mass does not rely on proliferation, other factors such as higher conversion rate of acinar cells to ADIP cells due to hyperglycemia during earlier stages of FAKi treatment and/or better granulation because of improved blood glucose levels post-FAKi treatment could be the potential mechanisms for enhanced β -cell mass recovery”.

3. The scRNAseq data (Fig 6) provides the interesting two acinar clusters. What percentage of the acinar cells are in each cluster? Does the percentage change after FAKi as might be expected if there has been the conversion to insulin producing cells? Also it would be stronger to also provide cluster 2 in the graphs in Fig6c

We have extensively modified the original Fig. 6 (new Fig. 7) and Extended Fig. 7 (new ExtFig.8). The dot plot in the new Fig. 7b shows the two acinar clusters with or without FAKi treatment, as recommended. With respect to the percentage of the acinar cells, 65% of acinar cells belong to cluster 1 in the control samples. This number is higher (72%) in FAKi treated mice. At the time of this study, we were not aware of the significance/characteristics of the peri-islet acinar cells. Therefore, we were purposefully less stringent during the islet purification steps, which allowed us having samples “contaminated” with acinar cells. These contaminating acinar cells would then (i) serve as reference when analyzing the ADIP cells, (ii) enable us to study the potential effects of FAKi on acinar cells. Given that the current ratio may simply reflect the differences in the number of “contaminating” acinar cells (which is more likely to be cluster 2), we do not feel confident to include this data in the current manuscript. Having said that, in our follow up bioinformatic study (anticipated submission summer 2024), we will certainly determine potential shifts between different cell types following FAKi treatment.

4. The TEM images have been greatly improved in quality. However, their interpretation seems inaccurate. The secretory granules are how one identifies islet cell types ultrastructurally. In Fig 7B and C, what are labeled as ADIPs have the characteristic granules of the glucagon-producing alpha cell and what are labeled as the alpha cells are likely delta cells.

We thank the reviewer for pointing out this error. We have corrected the labeling accordingly (new Fig. 7e, f). We do agree with the reviewer that in what was labeled as ADIP cells in the original Fig. 7b & c (new 7e, f), immature- and mature beta granules along with alpha and delta granules could be detected. We called these polyhormonal cells ADIP, because in day 1 post-Faki treated samples we could find zymogen granules in these cells. Given that TEM does not proof progenitor-progeny relationship, in the new version, we label these cells polyhormonal with (7e) or without (7c) zymogen. In addition, the immature- and mature beta granules or alpha and delta granules are highlighted in the revised version.

What is labelled z (zymogen granule) in the control beta cell (Fig 7b and 7c are probably autophagic vacuoles/autophagosomes?)

Unfortunately, it is not clear what cell the reviewer is referring to. In the original figure 7 (shown here), no zymogens were labeled in the beta cells. (a') Inset from (a) illustrating a peri-islet cell containing zymogen granules (z), and insulin granules (arrows). The “z” in 7b marks a zymogen in a polyhormonal cell.

We do not consider ourselves as experts in electron microscopy. However, according to the literature and as shown here, an autophagosome is an organelle surrounded by a double membrane structure (image taken from Yla-Anttila et al., 2009, DOI: 10.1016/S0076-6879(08)03610-0).

5. The methods are skimpy at best. While it is permissible to say as “previously described”, it is necessary to cite the reference. Under the TEM, what is MJK solution? Karnovsky’s fixative? Again usually one references Reynold’s lead citrate etc.

We are sorry for the brief nature of TEM methods. We are now providing a more detailed description in the methods section:

*“TEM was performed as previously described”¹³. Briefly, after fixation (2.5% glutaraldehyde, 2% PFA in 0.1 M sodium phosphate buffer), tissues were rinsed twice in 0.1 M monosodium phosphate for 30 minutes and then placed in 1% osmium tetroxide in water for 1 hour. Tissues were rinsed twice in deionized water. The samples were then dehydrated in serial concentrations of ethanol. Then tissues were placed were preinfiltrated in half resin/half propylene oxide overnight. The next day, tissues were infiltrated in 100% resin for 5 hours, and were then embedded with fresh resin and polymerized at 60°C overnight. The embedded tissues were sectioned with a Leica EM UC6 ultramicrotome. The sections were stained with 4% aqueous uranyl acetate for 30 minutes and 2 minutes in **0.2% lead citrate** in 0.1 N sodium hydroxide (NaOH). TEM imaging was performed using Thermo Scientific Ceta 16 MP camera”.*

Specific points;

While it is understandable to have few nonhuman primates, giving 4 data points and statistics on n=1 /cohort (Figure 5f) is not appropriate.

We do appreciate the valid point raised by the reviewer. Our intention was certainly not to mislead the readers, as it was stated in the corresponding figure legends that the beta-cell area graph was based on n=1 for each cohort. Nevertheless, we do acknowledge the reviewer’s concern and have accordingly modified the graph shown in Fig. 5f.

It is difficult to understand Fig 1H. There are many Insulin + cells (many nuclei) in a space that is only 20 um, so how is this considered extraislet and not an islet?

In this context, a pre-existing islet should have some infiltrating tomato⁺/insulin⁺ cells residing among otherwise tomato⁻/hormone⁺ cells. The cluster shown in the referenced figure (newExtFig. 1b) consists entirely of tomato⁺/insulin⁺ cells. Furthermore, we do not have data supporting the presence of other cell types (non-beta endocrine cells, endothelial, etc.) that together would have made a functioning islet. For this reason, we prefer calling it an extra-islet cluster. Whether such a cluster should be defined as a newly formed islet or just a cluster of insulin⁺ cells is outside the scope of the current study.

There is no legend for Fig 3b.

We thank the reviewer for pointing out this issue. A legend for Fig. 3b has been added.

Fig 4e. The AUC for IPGTT for STZ+FAKi and STZ+ Vehicle are not significantly different, so one cannot say there is improved glucose homeostasis as in figure legend title and p8 text?

We agree with the reviewer, and have modified the figure legend title accordingly:

*“Figure 4. FAKi treatment leads to **partial** improved blood glucose homeostasis and increased β -cell mass in mice following near total ablation of β -cells”.*

Fig 4H. How is it determined that this islet is neogenic?

The referenced figure has been omitted in the revised version.

Ext. Figure 4. It is unclear how the 3 replicating (Ki67+) cells outside the islet are determined to be acinar cells without immunostaining?

We have replaced ki67⁺ acinar cells with ki67⁺ cells in the corresponding figure legend.

“Arrows in (a) show ki-67⁺ cells next to a cluster of endocrine cells”.

Reviewer #2 (Remarks to the Author):

1. Regarding point 5, the original question was: "The findings described in extended figure 1 are confusing --Single endocrine cells next to an islet-- Islets are heterogeneous and it is unclear the significance of this observation. Were these purported occurrences more commonly seen in FAKi-treated mice than in controls? If so, the observation has to be carefully defined (as in "presence of peri-insular endocrine cells as determined by immunostaining of insulin") and quantified vs. controls. Otherwise, these are at most anecdotal observations.

The author states in the rebuttal:

“Single cells such as one shown in Extended Figure 1, was occasionally seen only in the FAKi treated mice. We have modified the text to address this issue”

However, the text remains:

“Another intriguing feature found in the pancreas of FAKi-treated mice was the appearance of clusters of endocrine-like cells within acini (Extended Fig. 1c, c”).

This still needs clarification. Since islets are naturally interspersed with acini, what makes a “cluster of endocrine-like cells” different from a small islet? Also, immunostaining for INS is still necessary. The reviewer understands that subsequent studies do show this. However, this is the first time that this is mentioned, so proof must be shown at this time, or at least the text modified accordingly.

We thank the reviewer for these comments. We have replaced the original ExtFig. 1 with a completely new figure (new Ext.Fig. 1). In addition, we provide corresponding immunostaining of insulin and amylase performed on the islet previously shown as ExtFig.1c (new Fig 1f, g). We have also modified the results accordingly.

Page 4 (Results section):

“The pancreas of FAKi-treated mice displayed overall normal histology. However, upon closer examination, we could occasionally find clusters of cells co-expressing insulin and amylase in the FAKi-treated pancreas (Extended Fig. 1a, a’). These insulin⁺/amylase⁺ cells (normally not detected in the wild type pancreas), resembled those found in the PdxCre;Fak^{fl/wt} pancreas²⁵ and displayed lower levels of insulin and amylase compared to the regular acinar or β-cells (Extended Fig. 1a, a”-a”). Anecdotally, we could also detect few single endocrine cells adjacent to islets (Fig. 1f). Presence of peri-insular endocrine cells was determined by immunostaining of insulin (Fig. 1g)”.

2. Regarding the reviewer’s comment that Extended figure 4 is very difficult to interpret: The original critique was that "according to Ext. Fig. 4b, no animals became diabetic. This is in contrast with the data shown in 4c. If insulin pellets last until week 5, why does glycemia not rise above 150 mg/dl in any group after that time point? Please clarify".

The author responds that “the earliest values (week 1) refer to the first readings after insulin pellet implantation. Therefore, the BG values are below 300. As demonstrated, once the pellet started to resolve (around week 5) the STZ-vehicle cohort displayed BG values above 300. We have modified the figure and the legend to clarify.” A corrected graphic is shown.

The reviewer appreciates the clarification. However, it is not appropriate to omit the readings prior to week one. The diabetic state needs to be shown. The green arrow points at some day between day 1 and 7, and by then we are told that the animals are already diabetic. The most widely accepted

standard for diabetic mice is three consecutive readings above 250 mg/dl. This needs to be shown. If the x axis unit is weeks, then it would be helpful to have a side chart corresponding to the interventions and readings happening during the first seven days. A supplementary table with the data would also be acceptable.

This data has been included in the new Fig 4c, as recommended by the reviewer.

It is also unfortunate that there is only one time point (week 6) where there is some meaningful degree of separation between the FAKi and vehicle STZ groups, and, given the error bars, it is probably not statistically significant. If the trend was towards separation, the authors must discuss why the experiment did not go on for longer, and at any rate state the p values corresponding to the statistical differences between groups.

The study was terminated on week 6 because mice in the STZ+vehicle cohort displayed signs of distress and we were asked to euthanize them. The p value has been added to this figure.

3. Regarding 12, the original critique was: These data indicate that like b-cells, ADIP cells display an insignificant proliferation rate under normal conditions, whereas they tend to expand more robustly in the background of β -cell ablation". Where are the BrdU data to substantiate this claim?

The authors respond: "The increased beta-cell mass is the result of proliferation (supporting data has been added). However, other factors such as higher conversion rate of acinar cells as due to hyperglycemia during earlier stages or better granulation because of improved blood glucose levels cannot be ruled out. We mention these issues in the Discussion."

In relation to this, the supporting data added by the authors are in Extended figure 4, where a single INS+/Ki67+ cell is shown. This is in no way the data that supports the above claim. If the authors believe that their data are consistent with beta-cell proliferation (rather than acinar-to-beta cell conversion, as in non-stz-treated mice), proper quantification needs to be shown. Ki67, not being cumulative, may not be the right tool to use: BrdU treatment to the live mice during the treatment would. In summary, as this claim is not supported by the presented data, it should be either removed or discussed as speculation.

We have performed the recommended BrdU study and have modified the text accordingly. Please, see our response to the Reviewer #1 (comment 2).

4. Regarding original critique 25: "These cells were more prevalent in the pancreas collected two weeks after FAKi, treatment (Fig. 5d, d'). This needs to be quantified."

The author states "We have rephrased this sentence".

However, the rephrasing is merely something along the lines of "the cells seemed more prevalent", which, frankly, looks like a rather easy way out of the reviewer's request. Any top journal (and this one is surely no exception) is very strong on rigor and statements being thoroughly backed by hard data. The use of expressions like "seemed" when applied to actual quantifiable data is not appropriate. The authors are encouraged to offer quantification or remove this statement altogether.

We have replaced the referenced statement with the following sentence:

Page 14 (Results section):

"In the pancreas collected ten days after FAKi treatment, no polyhormonal cells with zymogen granules could be found (Fig. 7f)".

Finally, it would be helpful for the reviewer to see the actual scRNAseq files, including FASTQ files and differential expression data by cluster, if available.

The reviewers have been granted access to the FASTQ files, as requested.

<https://www.ncbi.nlm.nih.gov/geo/query/acc.cgi?acc=GSE251852>

Token: **olebgmuohxiplmv**

Reviewer #3 (Remarks to the Author):

The authors present that treatment of adult mice and NHP with a small molecule that specifically inhibits kinase activity of focal adhesion kinase results in trans-differentiation of acinar cells into insulin producing β -like cells. It would be highly significant if the drug could increase beta cell volume and improves diabetes. However, there are several questions as to whether the data supports the conclusion.

Major comments:

1. Since pancreas transplants and islet transplants are being performed, the following statement in the abstract is inappropriate: “Insufficient functional β -cell mass causes diabetes; however, an effective cell replacement therapy for curing diabetes is currently not available”.

We have modified the introduction, as recommended.

Page 2 (Introduction):

“Thus, a cure for diabetes should entail replacement of insulin-producing β -cells. One approach is islet transplantation, a technique that during the past four decades has evolved into a routine clinical procedure with predictable efficacy for selected T1D patients¹. In addition to islet transplantation, there have been tremendous efforts throughout the years to generate new β -cells either through proliferation of pre-existing β -cells²⁻⁴ or by neogenesis using different sources such as embryonic stem cells⁵, duct cells⁶⁻¹¹, non- β -cells residing in the endocrine islets¹²⁻¹⁵, or acinar cells¹⁶⁻¹⁸”.

2. I think that the authors have not adequately addressed many of the issues raised by the reviewers in the last submission.

We believe most, if not all the issues raised by the reviewers have been adequately addressed in the revised version.

In addition, the number of individuals in animal experiments was generally small, so the basis for conclusions might be weak.

Low power by definition (In the absence of other biases, such as data selection) means that the chance of discovering effects that are genuinely true is low. Therefore, low-powered studies produce more false negatives than high-powered studies (PMID:23571845, PMID:21490505), and thus the detected statistical significance in the setting of low power is more likely to be real.

In the current study, we have shown the effects of FAKi treatment in converting acinar cells into insulin-producing cells in two animal species. Furthermore, our conclusions have been based on functional assays, different imaging methods, as well as transcriptomic analysis.

Lastly, the number of mice used in our studies (see below) was in average 5, which is acceptable for animal research.

Fig.1a-e: (n=7-10 for each cohort)

Fig.1h: (n=5 from 3 independent experiments)

Fig.1m-p: (n=5)

Fig.2 (n=5 from 3 independent experiments)

Fig.3a: (n=5 for each cohort)

Fig.3b (n=4)

Fig.3c,d: (n=5)

Fig.4a-e: (n=4-5 for each cohort)

Fig.4f,g: (n=3 per group), here β -cell quantification was performed on 3 out of 5 mice.

Fig.5 (n=4 for control and n=1 for FAKi-treated), **NHP** study

Fig.7: (n=3-5 for each cohort)

3. Why is the number of FAKi-treated NHPs one? If you are going to state scientific findings from this, please provide data from at least some individuals, I understand the difficulty of experimenting with NHPs, but I also understand the magnitude of variation on the other hand.

The NHP study was not planned in advance. The animal used in this experiment was originally intended to be part of an unrelated study (by other investigators). However, upon cancellation of that study, this “leftover” NHP was generously offered to our team. The low number of animals used in our NHP study remains as a limitation. We do acknowledge the exploratory nature of treating only one diabetic NHP with FAKi, and thus have toned down the conclusions of our NHP study.

Page 9 (Results):

“Encouraged by the effect of FAKi on mouse acinar cells and to further investigate the potential translational benefits of FAKi for diabetes treatment, in an exploratory study we investigated the effect of FAKi on a STZ-induced diabetic NHP”.

Page 10 (Results):

“Together, these findings warrant more rigorous and in-depth studies to determine whether FAKi treatment would lead to β -cell neogenesis in NHPs”.

Page 17 (Discussion):

“A limitation with the current study is that only one diabetic NHP was included in the FAKi-treated cohort. Nevertheless, the promising outcome of the current investigation should warrant for future studies involving larger cohorts of animals and more extensive analysis such as scRNA sequencing, c-peptide and HbA1c measurements or GTT. Together, this analysis would allow us to establish whether FAKi treatment converts acinar cells into ADIP cells in NHPs”.

Having said that, to our knowledge, there are no reports on spontaneous improvement of glucose homeostasis in STZ-treated diabetic NHPs. Therefore, in this context, a 60% reduction in exogenous insulin requirement, (despite the aforementioned limitation) should be considered as promising and worth reporting. Nevertheless, since the current study is otherwise conducted entirely in mice, the NHP data could be withdrawn, if needed.

4. Please show the blood glucose data for the NHPs.

Graphs showing the BG data for the NHPs have been added to Ext.Fig.6.

5. Please indicate the algorithm for determining the amount of insulin to be administered to the NHPs.

The following statement has been added to the methods section:

Page 1 (Methods):

“After induction of diabetes in the NHPs via IV STZ administration, NHPs received insulin therapy at a dose of ~ 1 unit/kg/day, 80% of the insulin dose came from long-acting insulin (Insulin glargine) divided into 2 doses, and 20% came from ultrashort-acting insulin (Insulin Lispro) that was given 10 minutes before meals (2 meals daily) according to a sliding scale. Blood glucoses were checked before meal 1 at 8 AM and before meal 2 at 5 PM. Insulin doses were titrated daily in view of the blood glucose to maintain blood glucose ~ 60-200 mg/dl”.

6. If ADIP cells are increased, is there a way to address type 1 diabetes autoimmunity?

This is indeed a very important question, which is currently being studied (funded by the JDRF and NIH). We have generated some very interesting data showing that ADIP cells are formed in FAKi-treated diabetic NOD mice. These mice show significant improved GTT compared to the vehicle-treated cohort up to three months after the completion of treatment. However, to what extent ADIP cells would avoid recognition by the diabetogenic T cells is the subject of our ongoing study. We hope to report these new findings soon.

7. Assuming an increase in insulin-producing cells in the animal, please indicate the blood insulin or C-peptide concentration, if possible.

The C-peptide measurements were performed by the pathology core lab. The results were unfortunately very inconsistent and many times below the detection threshold (surprisingly including the pre-STZ samples).

8. The effect of FAKi does not seem to be as great in mice or in NHP; extending the duration of FAKi administration or increasing the dosage should be considered.

The 30% recovery of the original beta-cell mass through neogenesis in STZ-treated mice may not be great, but it is a big leap forward. We thank the reviewer for this recommendation. However, while further optimization of the method is likely to be required, it is outside the scope of the current study.

9. The authors mentioned that the new ADIP cells will infiltrate into the islets, but how will they move from their original location, which pathway will they infiltrate from, and in which space will they settle to live?

The current manuscript is the result of near 8 years investigation on the use of FAKi treatment in converting acinar cells into insulin-producing cells. We acknowledge that the mechanism behind this phenomenon is not completely understood.

10. Is there a difference in the number and function of islets isolated from diabetic control and FAKi-treated mice?

Our quantification analysis entailed the beta-cell mass, and not the number of islets. Regarding the function we did not study isolated islets, as in our hands it is not feasible to isolate islets following near total ablation of beta-cells. Having said that, we believe that the partial improvement in GTT is the result of increased beta-cell mass and not individual cells function.

11. If possible, please describe glucose tolerance test data and HbA1c data from the NHP experiment.

We did not perform GTT nor have HbA1c data collected from the NHP experiment.

REVIEWER COMMENTS

Reviewer #1 (Remarks to the Author):

In their revision Dahiya et al added considerable detail and new data, but unfortunately their exciting and provocative conclusions need to be more rigorously supported to be convincing.

1. They have added images of BrdU incorporation (Fig2e) while extFig5 shows Ki67, but there is no quantification to support their statement that there is no increased proliferation that could account for the increased beta cells. Quantification of BrdU incorporation in control and experimental are needed.
2. They have added a few confocal images (the orthogonal view of z stack as shown re not very informative) but most findings need support with confocal images. Additionally it is a bit concerning that the same islet/image is repeated rather than showing other independent examples. The confocal image shown twice as Fig 2d and Ext Fig 2e is convincing but again a second islet would help strengthen the case. Fig1k and Fig7c are confocal images of the same field at different magnifications and settings as evident in the Black and white snapshot attached. Again multiple different images help support their case.
3. It needs to be clearly state in the paper itself what they consider as an islet. Most investigators in the field consider 6-8 endocrine cells clustered as an islet. In the rebuttal they state that since there were no other endocrine cell types in some of the clusters of insulin + cells, so they were not considered islets but rather as “extra islet clusters”.
4. While they have changed the nomenclature of the “mixed” cells seen by EM to polyhormonal, they still have erroneous identification of the cells and organelles. In Fig 7e the cells they describe as poly hormonal cells are glucagon producing alpha cells and what is labelled “z” in Fig7e is a secondary lysosome or late stage autophagosome. In the rebuttal they show a early stage autophagosome but there are many studies that show late stages often celled secondary lysosomes look like their “ zymogen granules” found in the endocrine cells.
5. It would be helpful for the reader to have information re the age and treatment of the mice within the figure legend.

Minor points:

1. There are discrepancies in the text : p5 line133 “we could not find any Tom+GCG+ cells in FAKi treated...” but Figure 1j legend states “ showing Tom+GCG+ cells within the islet”.
2. Unclear in STZ treated mice there were multiple blood glucose readings each week to give the “average weekly blood glucose level” as in Ext Fig 4.
3. Ext Fig 7. The untreated STZ diabetic NHP in B has much larger islets than either the naïve or the FAKi treated STZ diabetic NHP showing the huge variation within the NHP and why multiple animals are needed.
4. P4 line 121 “these Tom+Ins+ cells had little to no GLUT2”. To show a decreased level, there should be a comparison to what is seen in the intact preexisting islets or a control stained and imaged at the same time.
5. Ext Fig1. Area 1 is said to be a “cluster of amylase+ insulin + cells” , but there are no nuclei within the faintly green area so how be sure there are cells there?.

Reviewer #2 (Remarks to the Author):

The reviewer's concerns have been addressed. Some references to figures or extended figures in the rebuttal are mislabeled, but they are fine in the main text. While acknowledging that this manuscript still leaves many open questions, the significance of these initial findings is substantial enough to merit publication.

Reviewer #3 (Remarks to the Author):

The revised manuscript was greatly improved. However, regarding the NHP experiment, it is difficult to conclude that FAKi improved glucose metabolism based on the presented data alone.

General statement

We would like to thank all three reviewers for their comments. The revised manuscript is certainly an improved version of our previous submissions, mainly because of the feedback we received from the reviewers.

We agree with Rev#2 in that as exciting as these findings are, there are many questions to be answered. We hope to report our follow-up studies in the years to come.

Regarding Rev#3's comments on the NHP study, we are hopeful that publishing this report will open the door for fundings that would secure more in-depth and rigorous NHP studies.

Reviewer #1 (Remarks to the Author):

1. They have added images of BrdU incorporation (Fig2e) while extFig5 shows Ki67, but there is no quantification to support their statement that there is no increased proliferation that could account for the increased beta cells. Quantification of BrdU incorporation in control and experimental are needed.

Quantification of BrdU labeling has been added to figure 4 (Fig. 4i), as recommended by the reviewer.

2. They have added a few confocal images (the orthogonal view of z stack as shown re not very informative) but most findings need support with confocal images. Additionally it is a bit concerning that the same islet/image is repeated rather than showing other independent examples. The confocal image shown twice as Fig 2d and Ext Fig 2e is convincing but again a second islet would help strengthen the case. Fig1k and Fig7c are confocal images of the same field at different magnifications and settings as evident in the Black and white snapshot attached. Again multiple different images help support their case.

This manuscript contains a total of 15 images (including four confocal, Figs. 1k, 1l, 2d, 7c) showing tomato-red expression in conjunction with different endocrine markers in FAKi-treated non-diabetic or diabetic mice. In addition, we provide 2 similar images following conditional inactivation of gene encoding FAK. Moreover, the lineage tracing studies are backed up by scRNAseq (including pseudotime analysis), RNAscope and TEM. Not to forget, functional studies where we have observed partial glycemic improvement in FAKi-treated diabetic mice.

Having said that, while we have had many images from different mice showing tomato-red colocalize with insulin alone, the images used in Fig. 2d and Ext Fig. 2e were unique in the way that they showed three different cells in three different stages during conversion, within the same islet.

An orthogonal view of z-stacks is typically used to assure that the detected signals originate from the same cell(s). As such, we believe the presented images provide that information.

A confocal image of another islet has been added to replace Fig. 1i, as recommended by the reviewer.

3. It needs to be clearly state in the paper itself what they consider as an islet. Most investigators in the field consider 6-8 endocrine cells clustered as an islet. In the rebuttal they state that since there were no other endocrine cell types in some of the clusters of insulin + cells, so they were not considered islets but rather as "extra islet clusters".

The requested statement has been added to the discussion (page 15).

4. While they have changed the nomenclature of the “mixed” cells seen by EM to polyhormonal, they still have erroneous identification of the cells and organelles. In Fig 7e the cells they describe as poly hormonal cells are glucagon producing alpha cells.....

We respectfully beg to differ, as the polyhormonal cell shown in figure 7 contains granules typically found in α -, β -, or δ -cells, respectively.

Please, note the presence of immature β -granules (yellow arrowheads) in the polyhormonal cell. Similar granules can be found in a regular β -cell.

Polyhormonal cell in figure 7

Typical α - or β -cells in the wild type pancreas

...and what is labelled “z” in Fig7e is a secondary lysosome or late stage autophagosome. In the rebuttal they show a early stage autophagosome but there are many studies that show late stages often called secondary lysosomes look like their “ zymogen granules” found in the endocrine cells.

Secondary lysosomes in β -cells are shown in two recent reports. However, what is labeled as zymogen in our report does not show any resemblance with what seems to be considered as a typical secondary lysosome in the literature. Nevertheless, to address the reviewer’s concern we have removed the “z” labeling in the new figure 7e and modified the text accordingly.

Diabetologia (2021) 64:865-877 <https://doi.org/10.1007/s00125-021-05387-6>

Histochemistry and Cell Biology (2023) <https://doi.org/10.1007/s00418-023-02256-8>

Figure 7 “zymogen”

Hist & Cell (2023)

Diabetologia (2021)

5. It would be helpful for the reader to have information re the age and treatment of the mice within the figure legend.

The requested additional information has been added to the figure legend.

Minor points:

1. There are discrepancies in the text : p5 line133 “we could not find any Tom+GCG+ cells in FAKi treated...” but Figure 1j legend states “ showing Tom+GCG+ cells within the islet”.

A screenshot from the referenced figure legends shows otherwise. Nevertheless, we have looked for potential typos throughout the manuscript.

(i-j) Fluorescent imaging of tamoxifen-induced *ElaCreERT2,R26^{Tom}* mice treated with vehicle (i) or FAKi (j) for detection of tomato and glucagon showing Tom^+/Gcg^- cells within the islets. $n = 5$ mice from 3 independent experiments.

2. Unclear in STZ treated mice there were multiple blood glucose readings each week to give the “average weekly blood glucose level” as in Ext Fig 4.

The following sentence has been added to the figure legend for clarification:

“Random blood glucose was checked daily at the same time to calculate the average weekly BG. Graphs showing average weekly BG values:”

3. Ext Fig 7. The untreated STZ diabetic NHP in B has much larger islets than either the naïve or the FAKi treated STZ diabetic NHP showing the huge variation within the NHP and why multiple animals are needed.

The referenced image has been replaced with a new image that reflects the islet size variation in the naïve NHP pancreas.

4. P4 line 121 “these $Tom+Ins^+$ cells had little to no GLUT2”. To show a decreased level, there should be a comparison to what is seen in the intact preexisting islets or a control stained and imaged at the same time.

We thank the reviewer for highlighting this issue, as the way the text was formatted caused misunderstanding. The extra-islet Tom^+/Ins^+ that expressed $Glut2^+$ displayed no apparent differences compared to those within islets (regardless of tomato expression) (Fig.2b). However, the text was referring to the Tom^+/Ins^+ cells with no detectable $Glut2$ (marked with dotted line). We believe the $Tom^+/Ins^+/Glut2^+$ cells next to the $Tom^+/Ins^+/Glut2^-$ cells are exactly the kind of control that the reviewer was asking for.

We have modified Ext Fig2a and the text accordingly.

5. Ext Fig1. Area 1 is said to be a “cluster of amylase+ insulin + cells” , but there are no nuclei within the faintly green area so how be sure there are cells there?.

The faintly green area (yellow dots) is not amylase⁺. However, the cell next to it (pink dots) is insulin⁺/amylase⁺. Nevertheless, both cells appear to be Dapi⁺.

Reviewer #2 (Remarks to the Author):

The reviewer's concerns have been addressed. Some references to figures or extended figures in the rebuttal are mislabeled, but they are fine in the main text. While acknowledging that this manuscript still leaves many open questions, the significance of these initial findings is substantial enough to merit publication.

We thank the reviewer and could not agree more.

Reviewer #3 (Remarks to the Author):

The revised manuscript was greatly improved. However, regarding the NHP experiment, it is difficult to conclude that FAKi improved glucose metabolism based on the presented data alone.

We thank the reviewer for the feedback.

REVIEWERS' COMMENTS

Reviewer #1 (Remarks to the Author):

The new revisions have essentially addressed my critiques. However, there is still a lack of clarity re the BrdU incorporation:

1) how many islets in how many animals were counted? (this could be included in the Methods?Quantification analysis)

2) The BrdU is in the drinking water (changed daily) but the schema on Figure 4A does not include BrdU so it is not clear if BrdU is only during the 3 weeks during FAKi treatment or throughout the 10 weeks before harvest.

The authors should be congratulated on their exciting study, which now is well supported by their presentation.

Reviewer #1 (Remarks to the Author):

The new revisions have essentially addressed my critiques. However, there is still a lack of clarity re the BrdU incorporation:

1) how many islets in how many animals were counted? (this could be included in the Methods? Quantification analysis).

The requested information has been included in either the Methods section or the relevant figure legend.

Page 27 (Methods):

“To estimate the percentage of BrdU⁺/Ins⁺ cells in mice, whole pancreata were sectioned, and sections separated by 200-300 μ m were stained for insulin and BrdU. Data were obtained by analyzing approximately 70-80 islets per mouse”.

Page 42 (Figure legend 4i)

“(i) Quantification of proliferating insulin-expressing cells in ElaCreERT2,R26^{Tom} mice treated with STZ+vehicle or STZ+FAKi (n = 3 per group)”.

2) The BrdU is in the drinking water (changed daily) but the schema on Figure 4A does not include BrdU so it is not clear if BrdU is only during the 3 weeks during FAKi treatment or thought the 10 weeks before harvest.

Throughout this study, 2 experiments entailed BrdU administration, and on both occasions BrdU was administered in conjunction with FAKi treatment (3 weeks).

Occasion 1: In non-diabetic Ela^{Tom} mice (Figures 1 & 2 and Supplementary Figure 2).

Page 5 (Result section):

*“At eight weeks of age, these mice were then treated with FAKi or vehicle for three weeks, **in conjunction with BrdU administration via drinking water**. Finally, mice were sacrificed two weeks after cessation of the FAKi treatment”.*

Occasion 2: STZ-treated Ela^{Tom} mice (Figure 4h & i).

Page 9 (Result section):

*“To determine whether the partial β -cell mass recovery was due to higher proliferation rate among ADIP cells during FAKi treatment, diabetic Ela^{Tom} mice (high dose STZ) were then treated with FAKi or vehicle for three weeks, **during which they were given BrdU via drinking water. The mice were sacrificed immediately after cessation of the FAKi treatment.** The subsequent BrdU analyses showed no difference in proliferation between the two cohorts (Fig. 4h, i)”.*

Figure 4a refers to FAKi treatment of diabetic wild type mice, and did not entail BrdU incorporation. **The authors should be congratulated on their exciting study, which now is well supported by their presentation.**

We thank the reviewer for providing constructive comments and feedback throughout the peer review process.